# Phosphorylated HBO1 at UV irradiated sites is essential for nucleotide excision repair

Hiroyuki Niida[1], Ryoichi Matsunuma[1,2], Ryo Horiguchi[3], Chiharu Uchida[3], Yuka Nakazawa[4], Akira Motegi[5], Koji Nishimoto[1], Satoshi Sakai[1], Tatsuya Ohhata[1], Kyoko Kitagawa[1], Shinichi Moriwaki[6], Hideo Nishitani[7], Ayako Ui[8,9,10], Tomoo Ogi[11] & Masatoshi Kitagawa[1,12]

HBO1, a histone acetyl transferase, is a co-activator of DNA pre-replication complex formation. We recently reported that HBO1 is phosphorylated by ATM and/or ATR and binds to DDB2 after ultraviolet irradiation. Here, we show that phosphorylated HBO1 at cyclobutane pyrimidine dimer (CPD) sites mediates histone acetylation to facilitate recruitment of XPC at the damaged DNA sites. Furthermore, HBO1 facilitates accumulation of SNF2H and ACF1, an ATP-dependent chromatin remodelling complex, to CPD sites. Depletion of HBO1 inhibited repair of CPDs and sensitized cells to ultraviolet irradiation. However, depletion of HBO1 in cells derived from xeroderma pigmentosum patient complementation groups, XPE, XPC and XPA, did not lead to additional sensitivity towards ultraviolet irradiation. Our findings suggest that HBO1 acts in concert with SNF2H–ACF1 to make the chromosome structure more accessible to canonical nucleotide excision repair factors.

[1] Department of Molecular Biology, Hamamatsu University School of Medicine, 1-20-1 Handayama, Higashi-ku, Hamamatsu, Shizuoka 431-3192, Japan. [2] First Department of Surgery, Hamamatsu University School of Medicine, 1-20-1 Handayama, Higashi-ku, Hamamatsu, Shizuoka 431-3192, Japan. [3] Advanced Research Facilities and Services, Preeminent Medical Photonics Education and Research Center, Hamamatsu University School of Medicine, 1-20-1 Handayama, Higashi-ku, Hamamatsu, Shizuoka 431-3192, Japan. [4] Department of Genome Repair, Atomic Bomb Disease Institute, Nagasaki University, 1-12-4 Sakamoto, Nagasaki 852-8523, Japan. [5] Department of Radiation Genetics, Kyoto University Graduate School of Medicine, Kyoto 606-8501, Japan. [6] Department of Dermatology, Osaka Medical College, 2-7 Daigaku-machi, Takatsuki, Osaka 569-8686, Japan. [7] Graduate School of Life Science, University of Hyogo, 3-2-1 Kouto, Kamigori, Ako-gun, Hyogo 678-1297, Japan. [8] Department of Translational Oncology, St Marianna University, School of Medicine, 2-16-1 Sugao, Miyamae-Ku, Kawasaki 216-8511, Japan. [9] Institute of Medical Science, St Marianna University, School of Medicine, 2-16-1 Sugao, Miyamae-Ku, Kawasaki 216-8511, Japan. [10] Department of Genome Regulation and Molecular Pharmacogenomics, School of Bioscience and Biotechnology, Tokyo University of Technology, 1404-1 Katakuramachi, Hachioji, Tokyo 192-0983, Japan. [11] Department of Genetics, Research Institute of Environmental Medicine, Nagoya University, Furo-cho, Chikusa-ku, Nagoya 464-8601, Japan. [12] Laboratory Animal Facilities and Services, Preeminent Medical Photonics Education and Research Center, Hamamatsu University School of Medicine, 1-20-1 Handayama, Higashi-ku, Hamamatsu, Shizuoka 431-3192, Japan. Correspondence and requests for materials should be addressed to H.N. (email: niidah@hama-med.ac.jp).

Genome maintenance is vital for cell survival. Cyclobutane pyrimidine dimers (CPDs) and (6-4) photoproducts (6-4PPs) are the most common consequences of DNA damage following ultraviolet exposure. The nucleotide excision repair (NER) pathway repairs DNA damage and removes a wide range of DNA lesions—such as ultraviolet-induced CPDs and 6-4PPs—that distort stacking of the DNA double helix. NER is classified into two sub-pathways. Transcription-coupled repair (TC-NER) operates when transcriptional machinery comes into contact with a CPD or 6-4PP. Deficiency in TC-NER is associated with mutations in the CSA and CSB genes[1]. The second mechanism is global genome NER (GG-NER), which removes ultraviolet-induced DNA damage from both transcribed and non-transcribed DNA strands. A deficient GG-NER pathway results in xeroderma pigmentosum (XP). In eukaryote cells, removal of photolesions by GG-NER is initiated by binding of the XP complementation group C (XPC) protein[2,3]. Although XPC has a high affinity for 6-4PPs, its binding affinity for CPDs is rather weak because the distortion of the DNA structure by CPD is faint and thus efficient recognition of CPDs requires the presence of DDB2 (ref. 4). Cells derived from XPE patients that do not have DDB2 activity are deficient in CPD repair and show reduced 6-4PP repair[4–8].

In response to ultraviolet damage, CUL4A, DDB1, RBX1 and DDB2 are rapidly recruited to ultraviolet-induced lesions and form an active ubiquitin ligase complex[9–11]. Several proteins are ubiquitylated by the CRL4 and DDB2 complex upon ultraviolet exposure, including the core histone H2A (ref. 12), XPC (ref. 13) and DDB2 itself[9,13]. Lesion recognition may be further enhanced by the CRL4 and DDB2-mediated ubiquitylation of XPC, which increases XPC-binding affinity for DNA in vitro[13,14].

We recently discovered that HBO1, a member of the MYST family of histone acetyl transferases (HAT), interacts with DDB2 and is ubiquitylated by the CRL4 and DDB2 complex. Furthermore, ATM and/or ATR-dependent phosphorylation of HBO1 at Ser50 and Ser53 is required for the ubiquitylation of HBO1 (ref. 15). HBO1 was originally identified as an ORC-binding HAT[16]. Several studies have shown that HBO1 forms complexes with various partner proteins for various functions. When JADE-1 is a component of the HBO1 HAT complex, the HAT preferentially acetylates histone H4 Lys 5 and 12 (ref. 17). JADE-1 and HBO1 HAT also interacts with CDT1 and acetylates histone H4 around replication origins to facilitate pre-replication complex formation[18]. However, HBO1 complexes containing BRPF1, 2 and 3 subunits acetylate histone H3 Lys 14 (refs 19–22), and the acetylation of histone H3 Lys 14 by HBO1 and BRPF3 at the replication origin induces recruitment of CDC45 (ref. 22).

In addition to histone acetylation, ATP-dependent cooperative chromatin remodelling can also change chromatin structure. In mammals, there are four subfamilies of chromatin remodelling complexes: SWI/SNF, ISWI, CHD and INO80. The SWI/SNF complex was recently found to associate with XPC. BRG1, the catalytic subunit of SWI/SNF, interacts with DDB2 and affects the recruitment of XPC at the site of ultraviolet lesions[23]. In addition, PARP1 associated with DDB2 facilitates polyADP-ribosylation of ultraviolet-damaged chromatin and recruits ALC1 (ref. 24), which is an ATP-dependent chromatin remodeller but not part of any of the four chromatin remodelling complexes. The INO80 complex stimulates repair of ultraviolet damage through enhanced recruitment of NER factors[25]. These remodelling factors are involved in the GG-NER pathway. Human ISWI also accumulates at local ultraviolet damage sites and promotes recruitment of Cockayne syndrome B (CSB) protein to facilitate resolution to damage-stalled transcription[26].

Although we recently reported that HBO1 is mainly phosphorylated by ATR and interacts with DDB2 after ultraviolet irradiation[15], the involvement of HBO1 in the repair of ultraviolet-damaged DNA has not been evaluated. Here we evaluate the precise role of HBO1 in the repair of ultraviolet-damaged DNA.

## Results

**HBO1 is phosphorylated at CPD sites by ATR.** We recently showed that HBO1 is phosphorylated by ATM and ATR after ultraviolet irradiation. Phosphorylated HBO1 then interacts with DDB2 and is ubiquitylated for proteasome-dependent degradation 2 h after ultraviolet irradiation[15]. We examined whether phosphorylated HBO1 is present at ultraviolet-damaged sites. To detect HBO1 at ultraviolet-damaged sites, we performed ultraviolet irradiation through a membrane with micropores (local ultraviolet irradiation) in HeLa cells depleted for HBO1, DDB2 or XPC proteins (Supplementary Fig. 1a). Although accumulation of HBO1 was not detected in CPD-positive areas in shControl (shCtl) cells (Fig. 1a, upper panel), Ser50- and Ser53-phosphorylated HBO1 (HBO1 pS50/53) was co-localized with CPD 30 min after $12 \, \text{J m}^{-2}$ ultraviolet irradiation in shCtl cells (Fig. 1a, bottom panel).

Although ubiquitylation of HBO1 by DDB2 leading to degradation appears from 2 h after ultraviolet irradiation[15], HBO1 pS50/53 signals were detected on the chromatin at least 2 h after ultraviolet irradiation (Fig. 1b). During this period, phosphorylated HBO1 was not increased in the soluble fraction (Fig. 1b). Interestingly, the HBO1 pS50/53 signal was diminished in DDB2-depleted cells (Fig. 1a, bottom panel and graph, Supplementary Fig. 1b–d). In cells depleted for XPC, which is recruited to CPD sites by DDB2, HBO1 pS50/53 was slightly reduced at CPD sites (Fig. 1a, bottom panel and graph, Supplementary Fig. 1b–d).

To confirm whether the phosphorylated HBO1 detected at ultraviolet-damaged sites was ATM- or ATR-dependent, cells were treated with inhibitors against ATM and/or ATR before ultraviolet irradiation. We performed immunostaining for all proteins on MeOH/acetone-fixed cells except for CPD. MeOH/acetone treatment retains many proteins, including DDB2, XPC and HBO1 pS50/53, at damage sites, although sensitivity of detection was slightly less than CPD staining method. We also performed experiments using different ultraviolet doses and observed the same effects in our experiments (Supplementary Fig. 1e). Our results showed that cells treated with ATR inhibitor displayed marked reductions of phosphorylated HBO1 at DDB2-accumulated ultraviolet-damaged sites (Fig. 1c). This result indicates that ultraviolet-dependent phosphorylation of HBO1 is performed by ATR.

These results suggest that the reduction of pS50/53 in shXPC cells would be due to a defect of ATR activation at G1 phase in these cells, because activation of ATR in non-replicating or G1 cells is dependent on NER[27,28]. We examined this possibility by quantifying the fluorescent intensities of pS50/53 and DNA content in cells and using DNA contents to distinguish cells in G1 and S-G2/M phase (Fig. 1d). The intensity of pS50/53 at G1 phase in shXPC cells was reduced to 48% of that observed in shCtl cells. The pS50/53 intensities at S-G2/M in shXPC cells were almost the same as shCtl cells. Similar results were obtained in cells depleted for XPC using another XPC siRNA sequence (XPC-2) (Supplementary Fig. 1f). Consistent with previous reports[27,28], ultraviolet-induced phosphorylation of H2AX (γH2AX) in shXPC cells at G1 was much lower than control cells (Fig. 1d).

**HBO1 is required for GG-NER but is less involved in TC-NER.** Our results showed that phosphorylated HBO1 is present at ultraviolet-induced damage sites. To evaluate whether HBO1 is

involved in NER, we measured the ultraviolet-induced unscheduled DNA synthesis (UDS), which is an indicator of GG-NER capacity (Fig. 2a). To exclude the incorporation of 5-ethynyl-2′-

deoxyuridine (EdU) by DNA replication, shCtl, shDDB2 and shHBO1 cells were synchronized at M phase by nocodazole treatment and then released by washing with nocodazole and

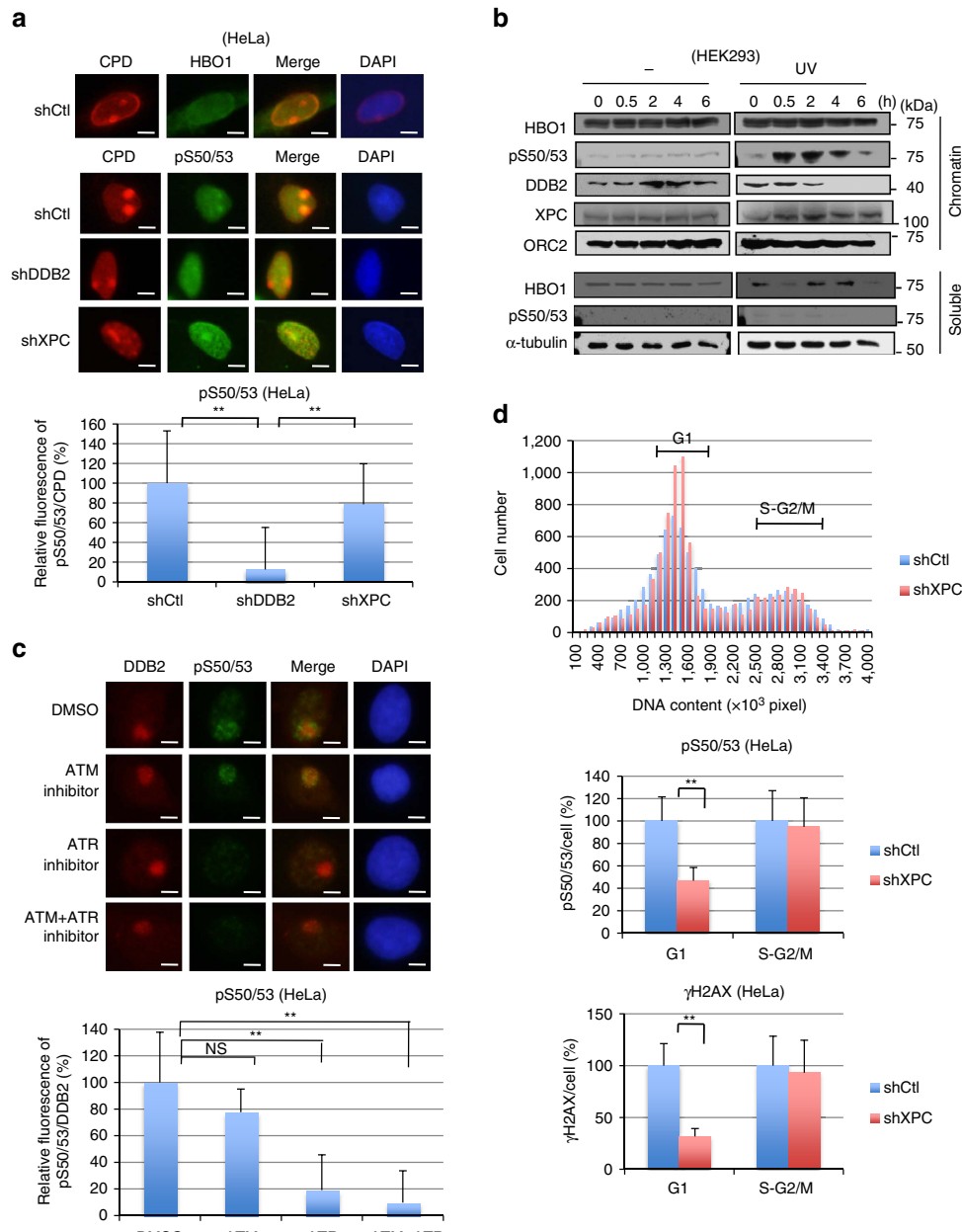

**Figure 1 | Phosphorylated HBO1 is detected in ultraviolet-induced DNA damage sites. (a)** Representative images of immunostaining for CPD and HBO1 or HBO1 pS50/53. HBO1 was phosphorylated at sites of local ultraviolet irradiation. shCtl, shDDB2 and shXPC cells were irradiated with 12 J m$^{-2}$ ultraviolet through a membrane with 5-µm pores. Cells were cultured for 30 min. Anti-HBO1 and anti-HBO1 pS50/53 antibodies were used for co-staining with anti-CPD antibody. The definition of relative fluorescence is shown in Supplementary Fig. 1b. The relative pS50/53/CPD intensities of shDDB2 and shXPC are shown as % of shCtl cells. Scale bars, 5 µm. n = 30 cells from three independent experiments; error bars indicate means ± s.d. **P < 0.01 (Student's t-test). Scanned images were subjected to linear adjustments of light–dark and contrast by Microsoft Power Point. (**b**) HBO1 pS50/53 is detected on the chromatin until 2 h after ultraviolet irradiation. HEK293 cells were mock-treated or irradiated with 50 J m$^{-2}$ ultraviolet. Cell lysates were fractionated into soluble and chromatin-enriched samples and then subjected to western blotting with the indicated antibodies. Scanned images were subjected to linear adjustments of lightã dark and contrast by Microsoft Power Point. (**c**) Inhibition of ATR reduced accumulation of HBO1 pS50/53 at the damage sites. Cells were incubated with ATM and/or ATR inhibitors for 2 h before local ultraviolet irradiation at 50 J m$^{-2}$. After 30 min, cells were fixed and immunostained with anti-DDB2 and anti-HBO1 pS50/53 antibodies. The relative pS50/53/DDB2 intensities of ATM and/or ATR inhibitor-treated cells are shown as % of DMSO control cells. Scale bars, 5 µm. n = 50 cells from three independent experiments; error bars indicate means ± s.d. **P < 0.01 (Student's t-test). (**d**) Phosphorylation of HBO1 Ser50 and Ser53 in G1 phase is dependent on XPC. Cells were irradiated 50 J m$^{-2}$ ultraviolet and cultured for 30 min. Cells were fixed and immunostained with anti-pS50/53 and anti-γH2AX antibodies. DNA was stained with DAPI. DNA contents and fluorescence intensities of pS50/53 or γH2AX in cells were quantified by the IN Cell Analyzer. Cells in G1 or S-G2/M phase were defined by DNA content. The average pS50/53 or γH2AX intensity in shXPC cells in G1 and S-G2/M phase is shown as % of shCtl cells. Cells (n = 1,500) from three independent experiments; error bars indicate means ± s.d. **P < 0.01 (Student's t-test).

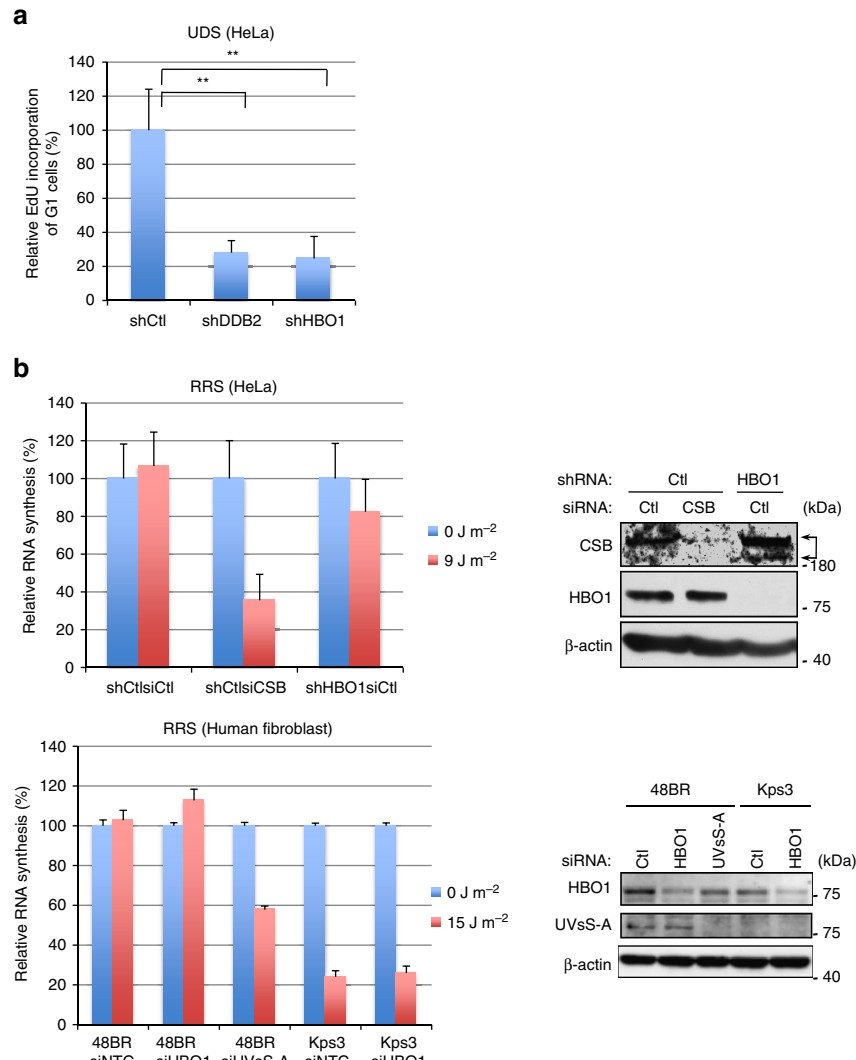

**Figure 2 | HBO1 is required for GG-NER but not TC-NER.** (**a**) Depletion of HBO1 caused defects of unscheduled DNA synthesis (UDS) in HeLa cells. Cells were irradiated with $15\,J\,m^{-2}$ ultraviolet and incubated for 3 h with EdU. Ultraviolet damage-dependent EdU incorporation of G1 cells was quantified and is shown as % of shCtl cells. Cells ($n = 1,500$) from nine independent experiments; error bars indicate means ± s.d. **$P < 0.01$ (Student's $t$-test). (**b**) Depletion of HBO1 did not affect recovery of RNA synthesis (RRS) after ultraviolet irradiation. HeLa shCtl or shHBO1 cells (left upper) and normal or UVsS-A fibroblasts (left bottom) were transfected with indicated siRNA. RRS assay was performed as described in Methods. Relative RNA synthesis is shown as % of mock irradiated. Depletion of CSB and HBO1 from HeLa and UVsS-A and HBO1 from fibroblasts was confirmed by western blotting (right). Cells ($n = 300$) from three independent experiments; error bars indicate means ± s.d.

cultured for 3 h. The majority of cells entered G1 phase, and then cells were irradiated with mock or $15\,J\,m^{-2}$ ultraviolet. After ultraviolet irradiation, cells were labelled with EdU for 3 h. To measure accurate EdU incorporation of G1 cells, total DNA contents were quantified by measuring fluorescence of Hoechst 33342 (Supplementary Fig. 2a) and EdU incorporation was quantified by the IN CELL Analyzer (Fig. 2a). Consistent with a previous report[29], ultraviolet damage-specific EdU incorporation of shDDB2 cells was reduced to 28% of control cells (Fig. 2a). EdU incorporation of shHBO1 cells was 25% of shCtl cells (Fig. 2a). These results indicated that HBO1 is involved in GG-NER.

Next, we performed recovery of RNA synthesis (RRS) assays to evaluate whether HBO1 is involved in TC-NER (Fig. 2b). Although depletion of CSB in HeLa cells caused a defect in RRS, HBO1-depleted cells showed almost the same level of RRS compared with controls (Fig. 2b, upper graph).

*UVSSA* is one of the causative genes of ultraviolet-sensitive syndrome (UVsS)[30], and the cells derived from UVsS-A patients

are defective in TC-NER. We also performed RRS assays in cells in which endogenous HBO1 or UVsS-A was depleted by siRNAs (Fig. 2b). In contrast to the reduction of RNA synthesis in UVsS-A-depleted cells after ultraviolet irradiation, cells depleted for HBO1 did not show any reduction of RNA synthesis after ultraviolet irradiation (Fig. 2b, bottom graph). Together these results indicated that HBO1 is required for GG-NER in the ultraviolet-induced DNA damage repair process.

**Functional HBO1 is required for XPC accumulation.** To examine the relationship between the presence of HBO1 pS50/53 and recruitment of XPC proteins to ultraviolet-damaged sites, we performed double immunostaining for DDB2 and XPC or XPC and HBO1 pS50/53 in cells depleted for DDB2, HBO1 or XPC (Fig. 3a, Supplementary Fig. 3a). Depletion of HBO1 eliminated pS50/53 signals (Supplementary Fig. 3a) and reduced the accumulation of XPC compared with mock-treated cells

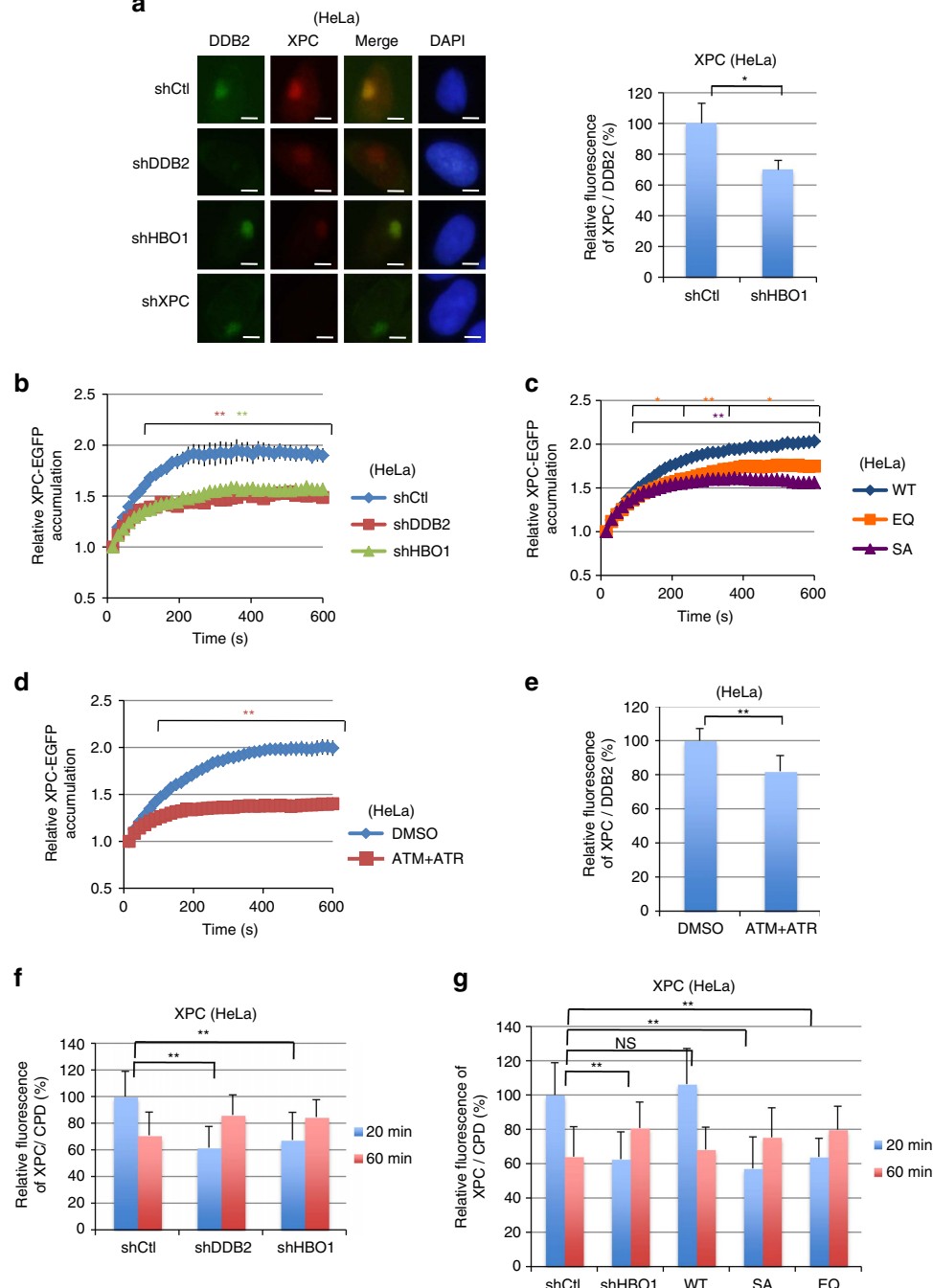

**Figure 3 | XPC accumulation is decreased in HBO1-depleted cells. (a)** Accumulation of XPC at damage sites was suppressed in shHBO1 cells. $50 \, \text{J m}^{-2}$ local ultraviolet-irradiated shCtl, shDDB2, shHBO1 and shXPC cells were incubated for 30 min and subjected to immunostaining with anti-DDB2 and anti-XPC antibodies. The relative XPC/DDB2 intensity of shHBO1 cell is shown as % of shCtl cells. Scale bars, 5 μm. Cells ($n = 50$) from three independent experiments; error bars indicate means ± s.d. *$P < 0.05$ (Student's $t$-test). **(b–d)** Live-cell imaging of XPC–EGFP accumulation at damage sites. Accumulation of XPC–EGFP was defective in shDDB2 cells and shHBO1 cells (**b**), shHBO1 cells stably expressing Myc-HBO1 EQ or SA (**c**), and ATM and ATR inhibitor-treated shCtl cells (**d**). XPC–EGFP was transiently transfected into shCtl, shDDB2 and shHBO1 cells and cells were irradiated with 405-nm laser in Hoechst 33342-containing medium. Quantification of accumulated XPC–EGFP was performed as described in Methods. Cells ($n = 30$) from three independent experiments; error bars are the mean ± s.e.m. *$P < 0.05$, **$P < 0.01$ (Student's $t$-test). **(e)** ATM and ATR inhibitor treatment suppressed XPC accumulation at ultraviolet-induced DNA damage sites. Cells were local irradiated with ultraviolet $50 \, \text{J m}^{-2}$ and cultured for 30 min. The relative XPC/DDB2 intensity of ATM and ATR inhibitor-treated cells is shown as % of DMSO control cells. Cells ($n = 50$) from three independent experiments; error bars indicate means ± s.d. **$P < 0.01$ (Student's $t$-test). **(f,g)** Relative XPC accumulation at damage sites at 20 and 60 min after $12 \, \text{J m}^{-2}$ ultraviolet irradiation was quantified and shown as % of shCtl cells after 20 min ultraviolet. Cells ($n = 50$) from three independent experiments; error bars indicate means ± s.d. **$P < 0.01$ (Student's $t$-test).

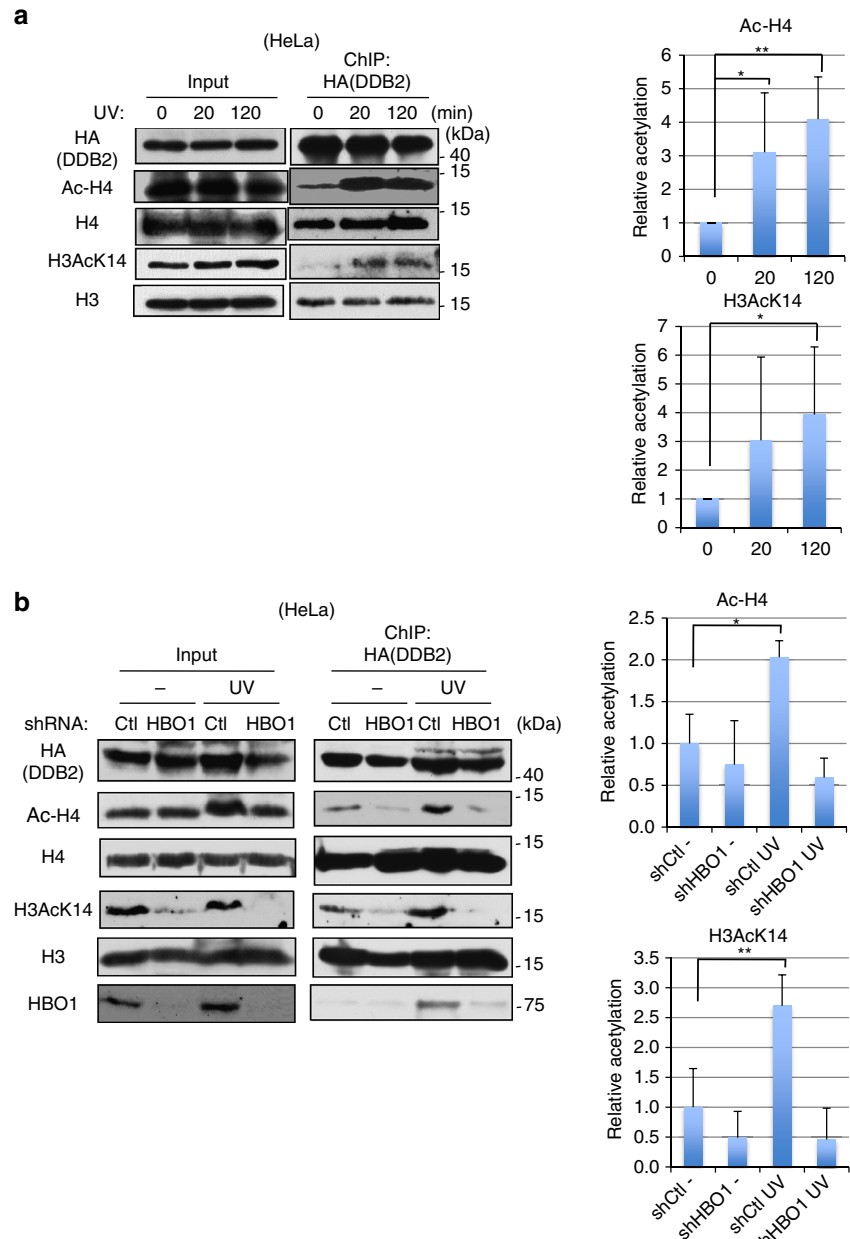

**Figure 4 | HBO1 acetylates histone H3 and H4 around DDB2-bound chromatin after ultraviolet irradiation.** (**a**) Acetylation levels of histone H3 Lys 14 and histone H4 around HA-DDB2-bound chromatin were increased after ultraviolet irradiation. Transiently expressing HA-DDB2 cells were irradiated with $40 \, \mathrm{J \, m^{-2}}$ ultraviolet and collected at 0, 20 and 120 min. Cells were cross-linked and immunoprecipitated as described in Methods. Immunoprecipitates were then subjected to western blotting with the indicated antibodies. Signal intensities were quantitated using ImageJ. Error bars indicate means ± s.d. from seven independent experiments. *$P < 0.05$; **$P < 0.01$ (Student's $t$-test). (**b**) Acetylation levels of histones around HA-DDB2-bound chromatin after ultraviolet irradiation were HBO1-dependently increased. HA-DDB2-expressing shCtl and shHBO1 cells were mock irradiated or irradiated with $40 \, \mathrm{J \, m^{-2}}$ ultraviolet and collected after 20 min ultraviolet irradiation. ChIP-western blotting was performed as described in **a**. Signal intensities were quantitated using ImageJ. Error bars indicate means ± s.d. from five independent experiments. *$P < 0.05$, **$P < 0.01$ (Student's $t$-test).

(Fig. 3a, graph). In addition to reduced accumulation of XPC in HBO1-depleted cells, accumulation of TFIIH p89, a further downstream factor in NER, was also suppressed in HBO1-depleted cells (Supplementary Fig. 3b). Thus, these results suggest that phosphorylated HBO1 enhances the accumulation of downstream factors in the GG-NER pathway.

Next, we examined XPC–EGFP accumulation kinetics at 405-nm laser-induced damage sites in living shHBO1 or shDDB2 cells. We first confirmed whether a combination of 405-nm laser and photosensitizer, Hoechst 33342 treatment, would generate

CPD and activate the NER pathway in HeLa cells. Although 405-nm laser irradiation combined with the photosensitizer Ro 19-8022 was reported to generate oxidative DNA damage that does not induce double strand breaks (DSBs) or CPD but accumulates XPC and CSB[31], our experimental conditions induced generation of CPD and accumulated GFP-XPA at the irradiated area (Supplementary Fig. 3c). In this condition, accumulation of XPC–EGFP in HBO1-depleted cells was suppressed to a level similar to that in DDB2-depleted cells (Fig. 3b, Supplementary Fig. 3d). To exclude the possibility of

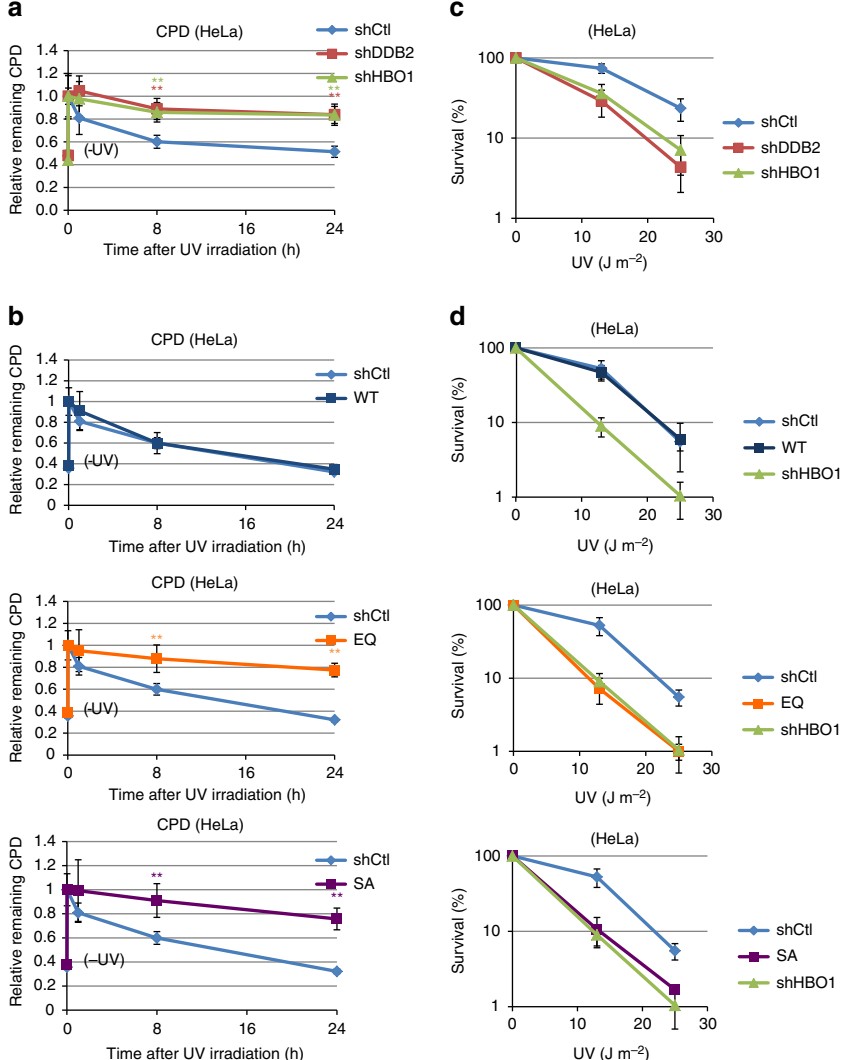

**Figure 5 | Depletion of HBO1 or mutant HBO1 causes defective CPD repair and increases sensitivity to ultraviolet irradiation.** (**a,b**) ShCtl, shDDB2 and shHBO1 (**a**) or shHBO1 Myc-HBO1 WT, EQ or SA-expressing cells (**b**) were not irradiated (-ultraviolet) or irradiated with 12 J m$^{-2}$ ultraviolet, and then cultured for 0, 1, 8 and 24 h. Cells were then fixed and immunostained with anti-CPD antibody and DNA was counterstained with DAPI. Relative intensities of CPD normalized against DAPI are shown as fold of ultraviolet 0 h cells. Cells ($n = 50$) from three independent experiments; error bars indicate means ± s.d. **$P < 0.01$ (Student's $t$-test). (**c,d**) ShCtl, shDDB2 and shHBO1 (**c**) or shHBO1 and shHBO1 Myc-HBO1 WT, EQ or SA-expressing cells (**d**) were irradiated at the indicated ultraviolet dose. Cells were cultured for 2 weeks and then fixed with MeOH/acetic acid and stained with crystal violet solution. The numbers of colonies were counted and are represented as % of mock-irradiated cells. Three independent experiments were performed. Error bars indicate means ± s.d.

off-target effects, we performed the same experiment using other shHBO1 and siDDB2 target sequences. Consistent with the initial results, depletion of endogenous HBO1 by shHBO1-2 or siDDB2-2 also suppressed XPC–EGFP recruitment at the DNA irradiated area (Supplementary Fig. 3e).

To confirm the importance of HBO1 phosphorylation by ATM and ATR and to examine whether the HAT catalytic activity of HBO1 is required for the accumulation of XPC at DNA damage sites, we monitored the accumulation of XPC–EGFP in cells stably expressing shHBO1-resistant Myc-tagged HBO1-WT (wild-type HBO1) or the mutants HBO1-EQ (Glu 508 substituted to Gln, catalytically dead) or HBO1-SA (Ser50 and Ser53 substituted to Ala) with concurrent shRNA depletion of endogenous HBO1 (Fig. 3c, Supplementary Fig. 3f,g). Expression of HBO1-WT completely restored the accumulation of XPC–EGFP in HBO1-depleted cells. However, HBO1-SA could not restore the accumulation of XPC–EGFP. Intriguingly,

the HBO1-EQ mutant partially restored the accumulation of XPC–EGFP at ultraviolet-damaged sites. This suggests that an additional function of HBO1 other than HAT activity might facilitate XPC loading at ultraviolet-induced damage sites.

We further demonstrated that inhibition of HBO1 phosphorylation by the ATM and ATR inhibitor also suppressed recruitment of XPC–EGFP at the irradiated area (Fig. 3d). Consistent with this defect, ATM and ATR inhibition reduced accumulation of endogenous XPC at ultraviolet-induced DNA damage sites (Fig. 3e). Suppression of recruitment of XPC would result in delayed accumulation of XPC at CPD sites. We therefore measured the relative quantity of XPC at CPD sites at a later time point (Fig. 3f). At 20 min after 12 J m$^{-2}$ local ultraviolet irradiation, the relative quantity of XPC at CPD-positive areas in shHBO1 and shDDB2 cells was 67% and 62% of shCtl cells, respectively (Fig. 3f). At 60 min after ultraviolet irradiation, XPC accumulation was reduced to 71% of levels observed in

shCtl cells at 20 min. On the other hand, XPC accumulation increased from 67% to 85% and from 62% to 86% in shHBO1 and shDDB2 cells, respectively (Fig. 3f). The delayed accumulations of XPC were also detected in HBO1-depleted cells stably expressing HBO1-EQ and -SA mutants (Fig. 3g).

Delayed accumulations of XPC at DNA damage sites in HBO1-EQ-expressing cells indicated that the acetylation activity of HBO1 is required for the completion of NER. Therefore, we next measured histone acetylation states around DDB2-accumulated CPD sites by performing ChIP-western blot assays using HA-tagged DDB2-transfected cells. Cells were ultraviolet irradiated with $40 \, \mathrm{J \, m^{-2}}$ and chromatin-bound DDB2 was immunoprecipitated and subjected to western blotting with anti-pan acetylated histone H4 (Ac-H4) and anti-acetylated histone H3 Lys 14 (H3AcK14) antibodies (Fig. 4a). Both H4 and Lys14 in H3 were acetylated in a time-dependent manner. To confirm that these events were dependent on HBO1, histone acetylation was measured in HBO1-depleted cells after ultraviolet irradiation (Fig. 4b). Consistent with our previous report[15], the interaction between HBO1 and HA-DDB2 was increased after ultraviolet treatment. Acetylations of H4 and Lys14 in H3 were significantly increased in shCtl cells (2.0- and 2.7-fold of the -ultraviolet control, respectively) (Fig. 4b). On the other hand, acetylations of histone H4 and H3K14 were not increased after ultraviolet irradiation in shHBO1 cells (Fig. 4b). Consistent with previous reports[22,32], HBO1 regulates H3AcK14 in the absence of ultraviolet damage. Based on these results, we concluded that ultraviolet-induced acetylation of H4 and H3 Lys14 around sites of ultraviolet lesions was HBO1-dependent.

**Functional HBO1 is involved in repair of CPDs and survival**. We demonstrated that HBO1 pS50/53 was present at CPD sites and that HBO1 acetylated H4 and H3 Lys14 at DDB2-bound chromatin (Figs 1a and 4). Therefore depletion or malfunction of HBO1 should delay the removal of CPDs or 6-4PPs. As expected, depletion of HBO1 delayed the removal of CPDs and 6-4PPs from chromatin similar to the delay observed after depletion of DDB2 (Fig. 5a, Supplementary Fig. 4a, respectively). Removal of CPDs and 6-4PPs in shCtl cells were detected at 1 h and 30 min after ultraviolet. Consistent with the delayed XPC accumulation at ultraviolet-irradiated sites in shHBO1 cells, removal of CPDs was significantly delayed in stable HBO1-depleted cells expressing shHBO1-resistant HBO1-EQ or -SA mutants (Fig. 5b).

Deficiency of CPD repair in HBO1-depleted cells and in cells expressing HBO1 mutants also elicited ultraviolet sensitivity in these cells (Fig. 5c,d). Depletion of HBO1 increased ultraviolet sensitivity similar to the depletion of DDB2 (Fig. 5c). Complementation experiments showed that only stable expression of HBO1-WT could rescue cell survival after ultraviolet exposure. Neither stable expression of HBO1-EQ nor -SA complemented endogenous HBO1 depletion (Fig. 5d). These results suggest that HBO1 is required for CPD repair and survival against ultraviolet irradiation.

We previously reported that Ser50 and Ser53 of HBO1 were phosphorylated in response to various DNA damage treatments[15]. Based on these observations, we examined whether HBO1 is involved in survival against ionizing radiation (IR), which induces oxidative damage, single-strand breaks (SSBs) and some DSBs, or mitomycin C (MMC) treatment, which induces both inter- and intrastrand crosslinks. We first monitored phosphorylation states of Ser50 and Ser53 in HBO1 (Supplementary Fig. 4b). Cells were irradiated with 10 Gy γ-irradiation or pre-treated with $50 \, \mathrm{ng \, ml^{-1}}$ MMC for 1 h and then washed. Cells were cultured for indicated times and

subjected to western blotting. Ser50 and Ser53 of HBO1 were immediately phosphorylated after γ-irradiation, and then phosphorylation was rapidly reduced (Supplementary Fig. 4b, left panel). MMC treatment induced and maintained phosphorylation of Ser50 and Ser53 of HBO1 until 6 h after removal of MMC (Supplementary Fig. 4b, right panel). Sensitivities of shHBO1 cells to IR and MMC were also examined (Supplementary Fig. 4c). The sensitivity of shHBO1 cells to IR was almost same as shCtl cells. Intriguingly, HBO1-depleted cells were sensitive to MMC treatment. These results suggested that HBO1 was required for not only ultraviolet but also MMC-induced DNA repair.

**Involvement of HBO1 in the canonical GG-NER pathway**. To evaluate whether the involvement of HBO1 in GG-NER observed in HeLa cells also occurs in human fibroblasts, we examined Ser50 and Ser53 phosphorylation of HBO1 at ultraviolet-induced damage sites in fibroblasts from normal and XP patient complementation groups XPE, XPC and XPA (Fig. 6a). Consistent with the depletion of DDB2 in HeLa cells, only 10% of phosphorylated HBO1 was detected at CPD sites in XPE fibroblasts that did not express DDB2. In XPC and XPA cells, the relative intensities of pS50/53 at CPD sites were 61% and 52% compared to levels detected in normal fibroblasts, respectively. PS50/53 signals were not observed in serum-starved (G0) XPA fibroblasts (XP7HM) (Supplementary Fig. 5a). The G1 population of XPA fibroblasts showed significantly lower pS50/53 signals after ultraviolet irradiation compared to normal fibroblasts (Supplementary Fig. 5b).

Next, we examined normal, XP and UVsS-A-deficient fibroblasts with stable HBO1 depletion (Supplementary Fig. 5c). XPE, XPC and XPA fibroblasts were deficient in the removal of CPDs compared with normal fibroblasts (Fig. 6b). In normal fibroblasts, almost 95% of CPDs were removed within 24 h after $12 \, \mathrm{J \, m^{-2}}$ ultraviolet exposure. However, in XPE, XPC and XPA fibroblasts, only 12%, 3% and 9% of CPDs were removed, respectively (Fig. 6b, blue lines). Further depletion of HBO1 in normal fibroblasts compromised CPD repair, but XP cells depleted for HBO1 did not show an additional delay in CPD removal (Fig. 6b, red lines).

Change in sensitivity towards ultraviolet irradiation was also evaluated in normal, XP and UVsS-A fibroblasts after depletion of HBO1 (Fig. 6c). Although normal fibroblasts became more sensitive after depletion of HBO1, XPE, XPC and XPA cells did not show additional ultraviolet sensitivity following depletion of HBO1. Consistent with UDS and RRS assays, depletion of HBO1 from UVsS-A fibroblasts induced additional ultraviolet sensitivity (Fig. 6c). A similar increased sensitivity was observed in CSB and HBO1 double-depleted cells (Supplementary Fig. 5d). Collectively, these results suggest that HBO1 remains at CPD sites where it facilitates canonical GG-NER in human fibroblasts.

**HBO1 recruits ACF1-SNF2H to ultraviolet-induced damage sites**. The tightly packed DNA in chromatin restricts access of repair proteins to sites of damage. To overcome such structural barriers, histone modifications, including acetylation and ATP-dependent chromatin remodelling, are required during NER[33]. Several reports have demonstrated an involvement of an ATP-dependent chromatin remodeller for NER[23-26,34]. We examined accumulations of BRM, BRG1, ACF1 and SNF2H at ultraviolet-irradiated areas by immunostaining after local-ultraviolet irradiation (Fig. 7a). Although BRG1 was implicated in mammalian NER[23], we could not detect its accumulation at DNA damage sites. However, consistent with previous reports[26,35], we could detect ACF1 and SNF2H accumulation at ultraviolet-damaged sites (Fig. 7a).

Both ACF1 and SNF2H are members of the ISWI chromatin remodelling family. ACF1 forms a complex with SNF2H, which is an ATPase catalytic subunit, and ACF1 and SNF2H are recruited to various DNA damage sites. For example, SNF2H interacts with XRCC1 to repair $H_2O_2$-induced DNA damage, which often generates SSBs and DSBs[36]. ACF1 and

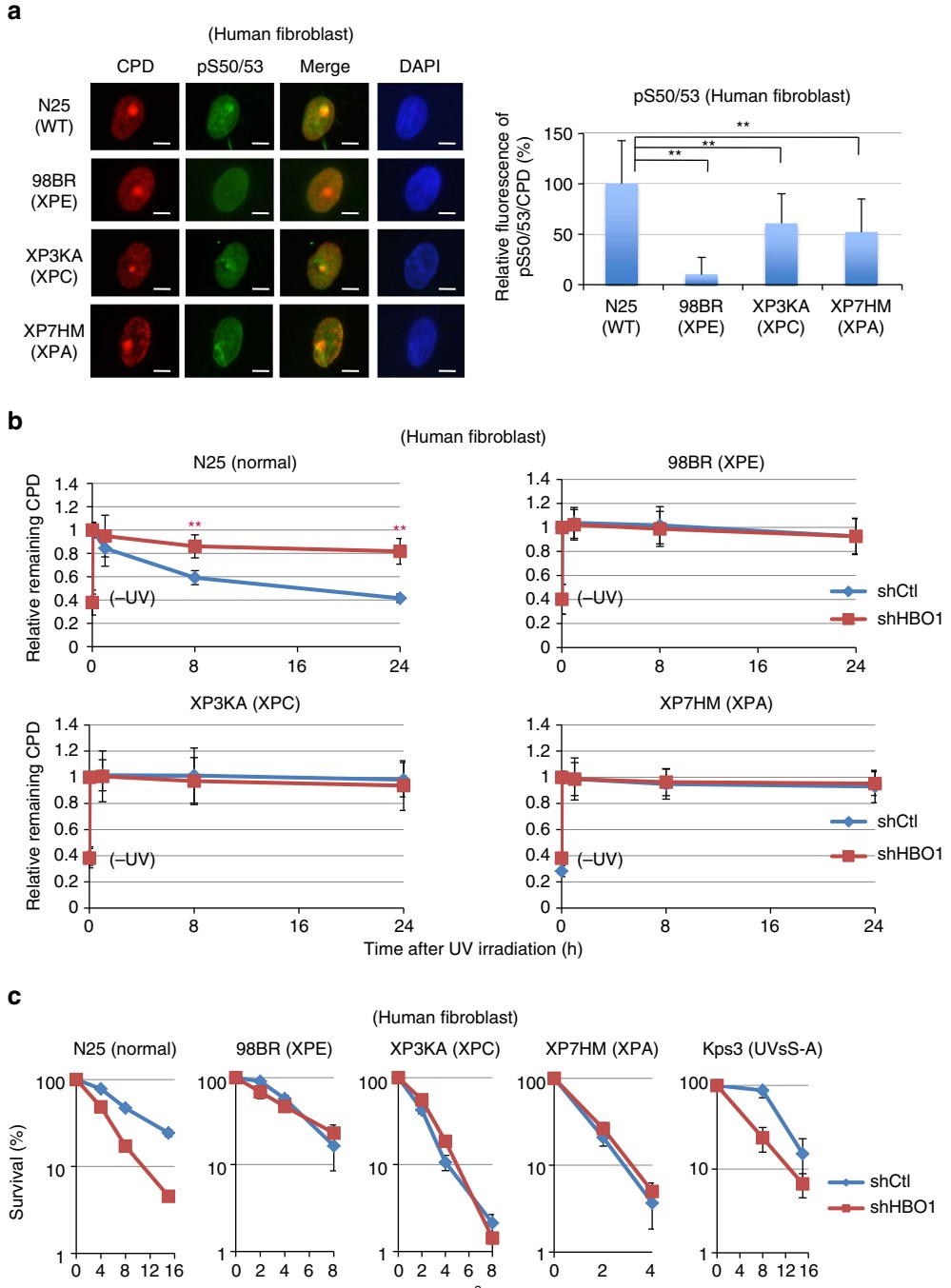

**Figure 6 | Phosphorylated Ser50 and Ser53 of HBO1 at sites of ultraviolet-induced damage is involved in canonical GG-NER pathway.** (**a**) Fibroblasts derived from normal and XPE, XPC and XPA patients were irradiated with $12\,J\,m^{-2}$ local ultraviolet and cultured for 30 min. Cells were immunostained with anti-CPD and anti-HBO1 pS50/53 antibodies. The relative pS50/53/CPD intensities of 98BR, XP3KA and XP7HM cells are shown as % of levels in N25 cells. Scale bars, $5\,\mu m$. Cells ($n = 30$) from three independent experiments; error bars indicate means ± s.d. **P < 0.01 (Student's t-test). (**b**) Fibroblasts derived from normal and XPE, XPC and XPA patients were depleted for HBO1 by a shHBO1-expressing lentivirus. Cells were not irradiated (-ultraviolet) or irradiated with $12\,J\,m^{-2}$ ultraviolet and cultured for 0, 1, 8 and 24 h, and then fixed and immunostained with anti-CPD antibody. Relative intensities of CPD normalized against DAPI are shown as fold of ultraviolet 0 h cells. Cells ($n = 50$) from three independent experiments; error bars indicate means ± s.d. **P < 0.01 (Student's t-test). (**c**) Fibroblasts derived from normal and XPE, XPC, XPA and UVsSA patients were depleted for HBO1 by a shHBO1-expressing lentivirus. Cells were irradiated at the indicated ultraviolet dose, cultured for 2 weeks and then fixed with MeOH/acetic acid and stained with crystal violet solution. The number of colonies were counted and are represented as % of mock-irradiated cells. Three independent experiments were performed. Error bars indicate means ± s.d.

SNF2H are recruited to irradiated sites by 405-nm laser with BrdU treatment, which also generates SSBs and DSBs, to operate the G2/M checkpoint[37]. ACF1 and SNF2H were also recently reported to accumulate at DSB sites to repair DNA by non-homologous end joining[38]. We confirmed that accumulation of ACF1 was not responsive to DSB induced by ultraviolet irradiation. Normal human dermal fibroblasts (NHDF) synchronized at G0 by serum starvation for 48 h

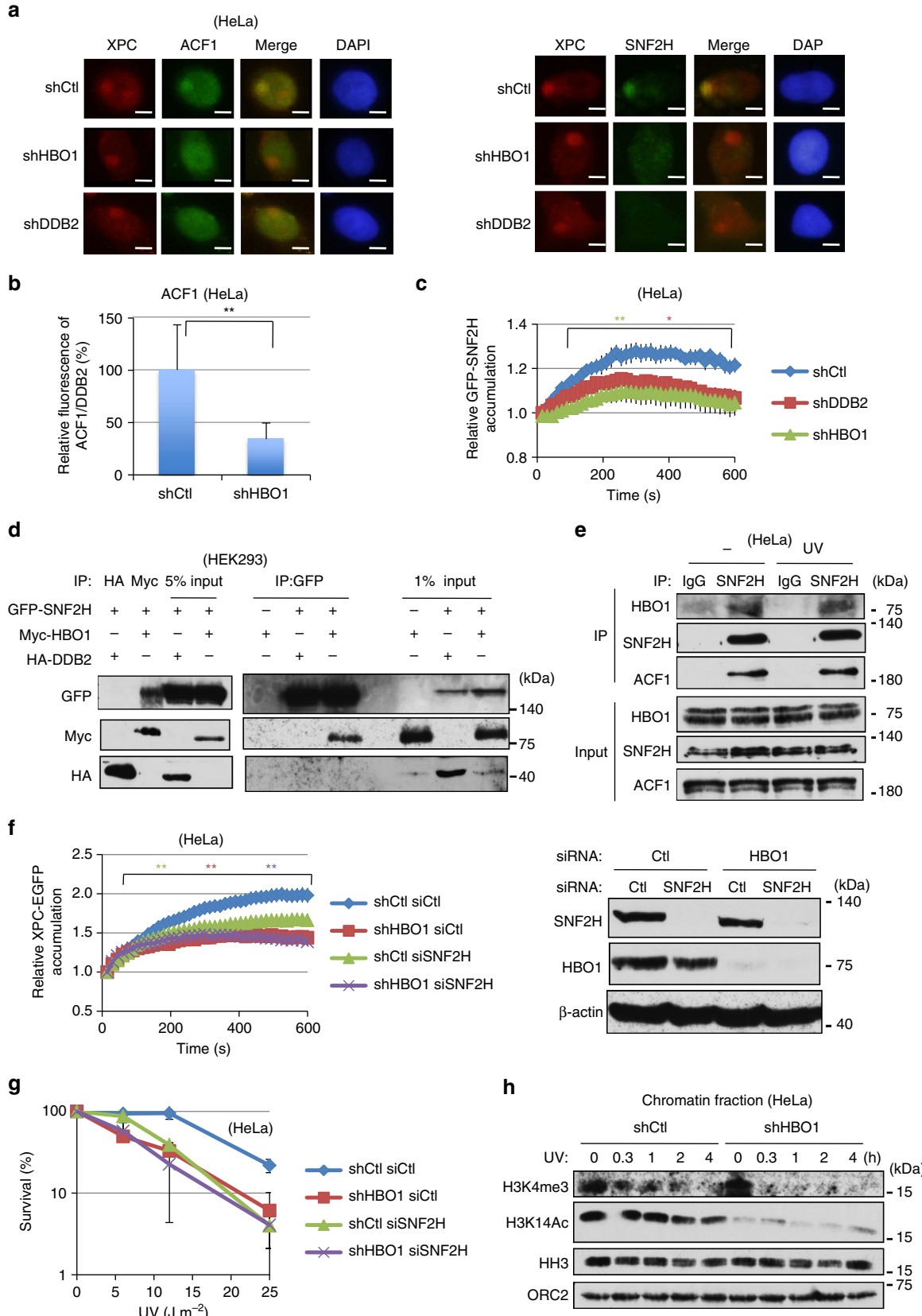

were irradiated with $50 \text{J m}^{-2}$ ultraviolet and after 30 min incubation, cells were immunostained with DDB2 and ACF1 (Supplementary Fig. 6a). About 70% of cells were arrested at G0 phase compared to 50% of asynchronized (AS) cells at G1. Both AS and G0 cells showed similar relative intensities of ACF1 (normalized to DDB2 levels) after ultraviolet irradiation (Supplementary Fig. 6a).

We also examined whether ACF1 and SNF2H accumulation involved DDB2 or HBO1 (Fig. 7a). Surprisingly, accumulations of both ACF1 and SNF2H at DNA damage sites were decreased in shDDB2 and shHBO1 cells (Fig. 7a,b, Supplementary Fig. 6b). We further examined SNF2H accumulation by monitoring GFP-SNF2H in living cells (Fig. 7c). Although 405-nm laser irradiation with Hoechst 33342 in the culture medium induces not only CPDs but also DSBs in DNA[39], we found reduced accumulation of GFP-SNF2H in shDDB2 and shHBO1 cells (Fig. 7c, Supplementary Fig. 6c). These observations suggest that DDB2 and HBO1 facilitate accumulation of the SNF2H complex at CPD sites.

To examine the molecular mechanism of ACF1–SNF2H accumulation at ultraviolet-induced DNA damage sites, we investigated whether HBO1 interacted with SNF2H. As shown in Fig. 7d, GFP-SNF2H successfully co-immunoprecipitated Myc-HBO1 but not HA-DDB2. However, we observed that despite high expression levels of Myc-HBO1, the amount of Myc-HBO1 that co-immunoprecipitated with GFP-SNF2H was low, suggesting that the interaction between GFP-SNF2H and Myc-HBO1 is not strong. Thus, we next evaluated the interaction between endogenous HBO1 and endogenous SNF2H. As shown in Fig. 7e, we found that endogenous HBO1 interacted with endogenous SNF2H in HeLa cells. These results suggest that HBO1 can interact with SNF2H in intact cells.

SNF2H was reported to recruit CSB at CPD sites in response to RNA polymerase II stacking[26]. The authors showed that α-amanitin, a RNA polymerase II inhibitor, diminished recruitment of SNF2H at ultraviolet-damaged sites and concluded that recruitment of SNF2H was transcription-dependent and it facilitated TC-NER. The authors also demonstrated that Trichostatin A (TSA), a histone deacetylase inhibitor, inhibited SNF2H recruitment at ultraviolet damage sites. To investigate the relevance between our observations and their report, we examined the effects of both treatments on HBO1 expression (Supplementary Fig. 7). Both α-amanitin and TSA treatment strongly suppressed HBO1 mRNA and protein expressions. Only about 30% of HBO1 protein was detected after 6 h of treatment with α-amanitin or TSA. Therefore, it is possible that at least part of the reduced SNF2H recruitment observed in previous reports[26] may be due to decreased HBO1 expression.

**HBO1 mediates recruitment of ACF1–SNF2H for GG-NER.** Accumulation of XPC–EGFP was only partially compromised in the HBO1-depleted cells expressing the HAT-inactive HBO1-EQ (Fig. 3c). This may indicate that HBO1-EQ can still recruit the ACF1–SNF2H complex at ultraviolet-induced DNA damage sites to facilitate XPC–EGFP loading. We next asked whether HBO1 is involved in ACF1–SNF2H-dependent facilitation of XPC loading on damaged DNA. We measured accumulation of XPC–EGFP in the ultraviolet-irradiated area of living cells depleted for HBO1, SNF2H or both HBO1 and SNF2H (Fig. 7f, Supplementary Fig. 8a,b). Live-cell imaging experiment showed that accumulation of XPC–EGFP was partially compromised in SNF2H-depleted cells compared with HBO1-depleted. To clarify whether depletion of SNF2H reduces XPC–EGFP accumulation, we repeated SNF2H depletion by siSNF2H-2 and examined its effect on XPC–EGFP accumulation (Supplementary Fig. 8b), and found that accumulation of XPC–EGFP was significantly decreased in SNF2H-depleted cells. In addition, double depletion of SNF2H and HBO1 did not show any additional defect in XPC–EGFP accumulation compared with HBO1 single depletion (Fig. 7f). We conclude that depletion of SNF2H decreases XPC accumulation at ultraviolet-induced damage sites.

Consistent with the XPC–EGFP accumulation experiments in living cells, depletion of HBO1 and depletion of both HBO1 and SNF2H resulted in higher ultraviolet sensitivity at ultraviolet $6 \text{J m}^{-2}$ compared with that of SNF2H single depleted cells (Fig. 7g). The yeast protein Isw1p and human SNF2H preferentially interacts with histone H3 tri-methylated Lys4 (H3K4me3)[40]. Therefore, we measured histone H3K4me3 modification after ultraviolet damage (Fig. 7h). Interestingly, histone H3K4me3 was immediately decreased in HBO1-depleted cells after ultraviolet irradiation. Collectively, these results indicate that HBO1, at least in part, interacts with and recruits ACF1–SNF2H at ultraviolet-induced DNA damage sites and would support even more ACF1–SNF2H interaction to chromatin by indirectly maintaining histone H3K4me3. Thus, HBO1 and ACF1–SNF2H cooperatively enhance XPC loading for efficient DNA repair of ultraviolet lesions by GG-NER.

## Discussion

XP cells have defects in genes involved in the NER pathway that are classified into seven different complementation groups, from XPA through XP-G (genes encoded by *XPA-XPG*), or defective DNA polymerase η, encoded by *POLH*[41–45].

Here we demonstrate that HBO1 is involved in the early step of CPD repair in GG-NER. Ser50 and Ser53 of HBO1 are phosphorylated by ATM and ATR in response to ultraviolet

**Figure 7 | The ACF1–SNF2H chromatin remodeller interacts with HBO1 and accumulates at ultraviolet-damaged DNA.** (**a**) ShCtl, shHBO1 and shDDB2 cells were irradiated with $50 \text{J m}^{-2}$ ultraviolet and cultured for 30 min. Cells were co-immunostained with indicated antibodies. Scale bars, 5 μm. (**b**) Depletion of HBO1-suppressed ACF1 accumulation at damage sites. ShCtl and shHBO1 cells were irradiated with $50 \text{J m}^{-2}$ ultraviolet and cultured for 30 min. Relative ACF1/DDB2 intensities of shHBO1 cells are shown as % of shCtl cells. Cells ($n = 50$) from three independent experiments; error bars indicate means ± s.d. **$P < 0.01$ (Student's $t$-test). (**c**) Live-cell imaging of GFP-SNF2H accumulation at damage sites. GFP-SNF2H was transiently transfected in shCtl, shDDB2 and shHBO1 cells. Laser irradiation was performed as described in Methods. Cells ($n = 30$) at least from three independent experiments; error bars are the mean ± s.e.m. *$P < 0.05$, **$P < 0.01$ (Student's $t$-test). (**d**) HEK293 cells overexpressing GFP-SNF2H and HA-DDB2 or Myc-HBO1 were immunoprecipitated with anti-HA antibody, anti-Myc-antibody and anti-GFP agarose. Immunoprecipitated samples were subjected to western blotting with the indicated antibodies. (**e**) Lysates from ultraviolet-irradiated or non-irradiated HeLa cells were immunoprecipitated with anti-SNF2H antibody. Immunoprecipitates were subjected to western blotting with the indicated antibodies. (**f**) Live-cell imaging of XPC–EGFP accumulation at damage sites. XPC–EGFP and siCtl or siSNF2H were transiently transfected in shCtl and shHBO1 cells. Laser irradiation was performed as described in Methods. Cells ($n = 30$) from three independent experiments; error bars are the mean ± s.e.m. **$P < 0.01$ (Student's $t$-test). (**g**) SiCtl or siSNF2H was transiently transfected in shCtl and shHBO1 cells. Cells were irradiated at indicated doses of ultraviolet. The numbers of colonies were counted and are represented as % survival of mock-irradiated cells. Three independent experiments were performed. Error bars indicate means ± s.d. (**h**) Histone H3 Lys 4 tri-methylation was rapidly decreased in shHBO1 cells after ultraviolet. ShCtl and shHBO1 cells were irradiated with $40 \text{J m}^{-2}$ ultraviolet and collected at indicated times. Chromatin-enriched fractions were subjected to western blotting using the indicated antibodies.

damage[15]. The ATM and ATR protein kinases are activated when genomic DNA suffers genotoxic stress and the activated kinases then phosphorylate downstream targets involved in cell cycle checkpoints, DNA repair, apoptosis and senescence. In general, these processes arrest cell proliferation until DNA repair is completed to suppress accumulations of mutations. Our current findings showed that phosphorylation of HBO1 Ser50 and Ser53 on chromatin was already at maximum levels at 0.5 until 2 h after ultraviolet irradiation (Fig. 1b), and then ubiquitylation of HBO1 by DDB2 occurs 2 h after ultraviolet treatment[15]. At 3 h after ultraviolet irradiation, the ubiquitylated HBO1 protein is degraded by the proteasome[15]. DDB2 also ubiquitylates itself after ultraviolet irradiation. In the current study, DDB2 disappeared from chromatin at 4 h after ultraviolet irradiation (Fig. 1b). In contrast to the decreasing levels of DDB2 and HBO1, XPC loading on chromatin was at maximum levels from 0.5 h after ultraviolet irradiation (Fig. 1b). We propose that ATR-dependently phosphorylated Ser50- and Ser53-HBO1 interacts with DDB2 at CPD sites on chromatin, and that HBO1 complexed with ACF1–SNF2H might acetylate neighbouring histones to facilitate remodelling of chromosomes to decrease chromatin compaction.

ACF1 and SNF2H are chromatin remodellers that accumulate at ultraviolet-induced damage[26,35] and DSB sites[38], respectively. SNF2H targeting to ultraviolet-C damage sites depends on transcription and histone modification to recruit CSB[26]. These cells showed defects in RRS but UDS was normal[26]. The authors concluded that SNF2H facilitated TC-NER. Upon generation of DSBs, ACF1 interacts with KU70 to accumulate at DSB sites. Previous studies showed that depletion of ACF1 and SNF2H sensitizes cells to genotoxic reagents that induce (single and double) strand breaks, oxidative damage, replication stress and helix-distorting DNA damage[26,38]. Here we demonstrated that SNF2H, the ATPase subunit of the ACF1 complex, bound to HBO1 and was facilitated to localize to sites of ultraviolet-induced damage. We found that depletion of HBO1 resulted in UDS defects but normal RRS was maintained. To understand the relationship between transcription or histone deacetylation-dependent and HBO1-dependent SNF2H recruitment at ultraviolet damage sites, we examined HBO1 protein and mRNA expression after α-amanitin and TSA treatment and found that both treatments strongly repressed HBO1 mRNA expression with a subsequent decrease of HBO1 protein to about 30% of levels observed in control cells. Therefore, the defects of SNF2H recruitment observed by α-amanitin and TSA treatments may be due to the reduction of HBO1 protein.

Regarding the contrasting results of UDS and RRS assays in HBO1- and SNF2H-depleted cells, we speculate that this is the result of HBO1-dependent and -independent SNF2H recruitments. The results of UDS assays in HBO1-depleted cells might indicate that acetylation of histones is more important for GG-NER than chromatin remodelling by ACF1–SNF2H. However, SNF2H would be supplied for a TC-NER HBO1-independent mechanism. ISWI binds near the transcription start site[46] and may facilitate an open chromatin structure for efficient CSB association[26]. In general, histones of transcriptionally active genes are acetylated[47], and therefore SNF2H might be able to scan and bind to target nucleosomes in the vicinity of lesions without HBO1.

In addition to the interaction between HBO1 and SNF2H, we observed that histone H3K4me3 was maintained after ultraviolet irradiation in shCtl cells compared with shHBO1 cells (Fig. 7h). Isw1p, the yeast homologue of ISW1, preferentially interacts with histone H3K4me3 (ref. 40). Furthermore, a *cis*-cross-talk between histone H3 Lys14 acetylation and H3 Lys4 tri-methylation in

yeast was identified[48]. As we, and others, reported previously, the lack of mammalian HBO1 significantly reduced histone H3 Lys14 acetylation. Collectively, these data indicate that histone H3 Lys14 acetylation may induce H3 Lys4 tri-methylation via a yet-unknown mechanism, and that ACF1–SNF2H may be stably maintained at damage sites to induce chromatin remodelling. An interaction between HBO1 and SNF2H, as well as a histone modification, including histone H3 Lys14 acetylation by HBO1 and H3 Lys4 tri-methylation, which would be enhanced by histone H3 Lys14 acetylation, could support the localization of ACF1–SNF2H. The latter mechanism would be reasonable because in contrast to ACF1–SNF2H protein that accumulated at DNA damage sites, HBO1 protein amounts did not significantly change at chromatin sites in response to DNA damage. As HBO1 protein is already abundant on the chromatin, we speculate that HBO1 that is already present on chromatin is phosphorylated by ATR in response to ultraviolet-induced DNA damage. Although some HBO1 would already bind with SNF2H, additional ACF1–SNF2H would likely accumulate via histone H3K4me3 modification.

These findings raise another question—that is, if cells are in G1 phase of the cell cycle, how is ATR activated to phosphorylate HBO1? We speculate that ATR would be activated by the first NER, that is, involving processing of the lesion by NER and the damage signalling activating ATR owing to the resulting NER-dependent single-strand break or single-strand gap induced by the dual incision. The involvement of NER in the phosphorylation of H2AX in non-dividing cells was reported previously[27]. Consistent with this idea, we detected significantly decreased phosphorylations of Ser50 and Ser53 in serum-starved XPA fibroblasts (Supplementary Fig. 5b). On the other hand, we did not detect a significant reduction in HBO1 phosphorylation in S-G2/M shXPC and XPA cells. However, replication-independent activation of ATR requiring NER was previously reported[49]. We speculate that replication-dependent ATR activation is dominant in S-phase cells and these cells accumulate in the G2 phase at the G2 checkpoint. Lack of replication-independent ATR activation in shXPC or XPA cells could not be detected. Thus, activated ATR would phosphorylate HBO1 around DNA damage sites. HBO1-dependent histone acetylation and accumulation of ACF1–SNF2H complex would disrupt the contact between damaged DNA and histones to facilitate access of XPC as well as downstream NER repair molecules, such as the TFIIH complex. This is consistent with our observation that HBO1-depleted cells showed increased sensitivity for MMC, because MMC-induced interstrand crosslinks require the NER pathway to repair these types of DNA lesions[50]. Our findings provide important evidence of a mechanism involving HBO1 and ACF1–SNF2H to optimize chromatin structure in the GG-NER pathway.

## Methods

**Plasmids and transfections.** The Myc-HBO1 SA mutant with serine/alanine substitutions at residues 50 and 53 and the catalytically dead Myc-HBO1 EQ mutant with glutamic acid/glutamine substitution at residue 508 were generated by PCR-based site-directed mutagenesis. HBO1 WT, EQ and SA mutants were subcloned into pCMV-Myc (Clontech). XPC was generated by PCR and subcloned into the pEGFP-N1 plasmid. pEGFP-SNF2H was a gift from Dr Yasui. Plasmid transfection was performed using X-tremeGENE 9 DNA reagents (Roche).

**Antibodies.** The antibodies used in this study were anti-Myc antibody (sc-40, Santa Cruz), anti-HA antibody (sc-57592, Santa Cruz), anti-HBO1 antibody (sc-13284, Santa Cruz), anti-DDB2 antibody (sc-81246, Santa Cruz), anti-XPC antibody (sc-30156, Santa Cruz), anti-TFIIH p89 antibody (sc-293, Santa Cruz), anti-CPD antibody (NMDND001, Cosmo Bio), anti-Orc2 antibody (sc-13238, Santa Cruz), anti-acetyl-Histone H4 antibody (#06-866 Millipore), anti-Histone H4 antibody (ab10158, Abcam), anti-acetyl-Histone H3K14 antibody (A-4023, EPIGENTEK), anti-Histone H3 (tri methyl K4) antibody (ab1012, Abcam),

anti-Histone H3 antibody (39763, ACTIVE MOTIF), anti-γH2AX antibody (50-171-736, Upstate), anti-SNF2H antibody (ab3749, Abcam), anti-ACF1 antibody (NB100-61041, Novusbio), anti-CSB antibody (553C5a, BIO MATRIX RESEARCH), anti-UVsS-A antibody (H00057654-B01, Abnova) and anti-β-actin antibody (sc-47778, Santa Cruz). Phospho Ser50 and Ser53 HBO1 rabbit polyclonal antibodies were generated by immunization with a synthetic phosphopeptide (CSARLpSQSpSQD). To perform immunoprecipitation of GFP-SNF2H, anti-GFP mAb-Agarose (D153-8, MBL) was used.

**Cell culture.** Human HEK293 and HeLa cells obtained from American Type Culture Collection (ATCC), primary normal human fibroblasts (N25 and NHDF), primary XPE fibroblasts (98BR), primary XPC fibroblasts (XP 3KA), primary XPA fibroblasts (XP 7HM) and primary UVsS-A fibroblasts were used. All cells were cultured in Dulbecco's modified Eagle's medium (DMEM) supplemented with 10% fetal bovine serum. For G0/G1 cell synchronization experiments, human fibroblasts were deprived of serum in growth medium containing 0.2% serum for 48 h.

**RNA interference.** Control, HBO1, DDB2 and XPC shRNA plasmids were purchased from Santa Cruz (sc-108065, sc-35530B-SH, sc-37799-SH and sc-37805-SH). The shRNA sequences used in this study are as follows: shHBO1-2, 5′-AAG CCCTTCAGATGCTCAAGT-3′; shControl-2, AAGAGGAGCATATTGGGA-AGA; siSNF2H, 5′-CGCUCUCCAUCGUAUGUAAAAUU-3′; SNF2H-2, 5′-CCGGGCAAAUAGAUUCGAGUAUUUA-3′; siDDB2-2, 5′-UCACUGGGCU-GAAGUUUAA-3′; siXPC–2, 5′-UAGCAAAUGGCUUCUAUCGAA-3′; siCSB, 5′-GCAGUAACUUCUAAUCGAAUU-3′; and siCSB-2, 5′-GAAGAGUUGUCA-GUGAUUA-3′. siRNA oligonucleotides targeting UVsS-A was purchased from Nippon EGT (sequences can be obtained from the company's website).

HeLa cells were transfected with HBO1 shRNA plasmid DNA and DDB2 shRNA plasmid DNA using X-tremeGENE 9 DNA reagents (Roche), in accordance with the manufacturer's protocol. Primary fibroblasts (normal, XPE, XPC, XPA and UVsS-A) were infected with Control shRNA Lentiviral Particles (sc-108080, Santa Cruz) or HBO1 shRNA Lentiviral Particles (sc-35530-v, Santa Cruz), in accordance with the manufacturer's protocol.

**Local ultraviolet irradiation.** Cells were cultured in 35-mm dishes for 24 h. Cells were then washed twice with PBS and covered with a polycarbonate isopore membrane filter (pore size, 5 μm or 8 μm). The covered cells were then irradiated with ultraviolet using five low-pressure mercury lamps (GL-10, Toshiba, Tokyo; predominantly 254 nm ultraviolet) at a dose rate of $0.43\,J\,m^{-2}\,s^{-1}$. Immediately after exposure, DMEM supplemented with 10% fetal bovine serum was added and cells were cultured at 37 °C for the indicated time before fixation.

**Immunofluorescence labelling.** For anti-CPD antibody staining, cells were triton-extracted with a buffer containing 0.2% Triton-X 100, 300 mM sucrose in PBS for 30 s and then fixed in 1% formaldehyde, 0.2% Triton-X 100, 300 mM sucrose in PBS for 20 min. Cells were blocked with PBS containing 10% fetal calf serum for 1 h and subsequently incubated with the primary anti-XPC antibody or anti-pS50/53 antibody diluted 1:100 in PBST (0.05% Tween 20) for 2 h, and then extensively washed with PBST. For concomitant detection of ultraviolet photolesions, cells stained with the indicated antibodies were fixed in 4% formaldehyde in PBS for 20 min, followed by denaturation of DNA in 5 N HCl for 5 min and extensive washing. The cells were then stained with anti-CPD antibody (1:2,000 dilution in PBST) and then incubated for 1 h with DAPI (1 ng ml⁻¹) and secondary antibodies conjugated with Alexa Fluor 488 or 594 fluorescent dyes (Molecular Probes, 1:500 dilution in PBST). After extensive washing, cells were mounted with VECTASHIELD mounting medium (Vector Laboratories). For staining with other primary antibodies, cells were fixed with methanol:acetone 1:1 solution for 10 min and then washed with PBS three times. Cells were blocked with PBS containing 10% FBS for 30 min and then incubated with primary antibody for 1 h. Cells were then washed with PBS twice and secondary antibodies conjugated with Alexa Fluor 488 or 594 fluorescent dyes were applied for 30 min. After extensive washing, cells were mounted with VECTASHIELD mounting medium.

**Quantification of phosphorylation in the cell cycle.** Cells immunostained with anti-pS50/53 or γH2AX antibodies were counterstained with DAPI to quantify DNA contents. Cells were gated as G1 or S-G2/M by DNA contents and intensities of pS50/53 or γH2AX were quantified in each cell by the IN CELL Analyzer 2200 (GE Healthcare Life Sciences).

**Unscheduled DNA synthesis.** Fluorescent-based UDS was performed as follows. Cells seeded on coverslips were arrested at M phase by 50 nM nocodazole for 15 h, released by washing with PBS and then cultured for 3 h. Cells were then ultraviolet irradiated with 0 or 15 J m⁻² and incubated for 3 h in medium containing 5-ethynyl-20-deoxyuridine (EdU; Invitrogen), followed by washing with PBS and fixation with 4% formaldehyde. Cells were permeabilized with 0.5% Triton in PBS, and EdU incorporation and total DNA were visualized using Click-it Alexa Fluor

594 and Hoechst 33342, respectively, according to the manufacturer's instructions (Invitrogen). The total DNA content of each cell was quantified by measuring Hoechst 33342. The specific repair capacity of G1 cells was quantified in 1,500 cells by determining the overall nuclear fluorescence of EdU using the IN CELL Analyzer 2200 (GE Healthcare Life Sciences). The fluorescence values were normalized to the fluorescence in shCtl cells, which was set at 100%.

**Recovery of RNA synthesis assay.** HeLa shCtl cells were transiently transfected with siControl and siCSB. ShHBO1 cells were transiently transfected with siControl. Normal human 48BR fibroblasts were transiently transfected with siControl, siHBO1 or siUVsS-A. UVsS-A Kps3 fibroblasts were transiently transfected with siControl or siHBO1. At 48 h after transfection, cells were subjected to RRS assays were performed. In brief, HeLa and fibroblast cells were irradiated with 9 and 12 J m⁻² ultraviolet, cultured for 12 h, and cells labelled with 5-ethynyluridine (EU) for 2 h. Incorporation of EU was visualized using Click-it Alexa Fluor 594 or Fluor 488 according to the manufacturer's instructions (Invitrogen).

**CPD repair assay.** Cells were irradiated with 12 J m⁻² ultraviolet, and then fixed and immunostained with anti-CPD antibody. DNA was counterstained with DAPI. Relative intensities of CPD were normalized to DAPI at indicated time points.

**Live-cell confocal laser-scanning microscopy.** For local ultraviolet irradiation of living cells, a Leica TCS SP8 laser-scanning confocal microscope was used. Cells were grown on a 35-mm glass-based dish (IWAKI, 3971-035) in medium containing 5 μg ml⁻¹ Hoechst 33342 at 37 °C and 5% CO₂. A rectangular region of interest (ROI) of a nucleus was irradiated five times with a × 63 lens at 60 Hz, zoom 2.0, 100% of 405-nm laser power. XPC–GFP or GFP-SNF2H accumulation was monitored with 0.2% of 488-nm laser power at 15-s intervals for 10 min. Images were acquired using LAS X 1.8 software (Leica). The mean intensity of each accumulated area was obtained after subtraction of the background intensity in the irradiated cell, and mean values of at least three independent experiments are given.

**Clonogenic survival assay.** For clonogenic assays, single-cell suspensions were generated for each cell line and specified number of cells were seeded into tissue culture plates. Cells were allowed to adhere for 16 h and then exposed to ultraviolet. At 2–3 weeks after seeding, depending on each individual cell line, colonies were stained with crystal violet. Colonies were counted to determine the surviving fraction. Surviving fractions were normalized to the plating efficiency of each cell line.

**Subcellular fractionation.** Subcellular fractionation of mammalian cells was performed. In brief, $6 \times 10^6$ cells were suspended in 400 μl of solution A (10 mM HEPES pH 7.9, 10 mM KCl, 1.5 mM MgCl₂, 0.34 M sucrose, 10% glycerol, 1 mM dithiothreitol, protease and phosphatase inhibitors), and Triton X-100 was added to a final concentration of 0.1%. The cells were incubated on ice for 5 min, and cytoplasmic and nuclear fractions were collected by centrifugation at 1,300g for 4 min. The isolated nuclei were then washed in solution A, lysed in 400 μl solution B (3 mM EDTA, 0.2 mM EGTA, 1 mM DTT, protease and phosphatase inhibitors), and incubated on ice for 10 min. The soluble nuclear and chromatin fractions were collected by centrifugation at 1,700g for 4 min. The isolated chromatin was then washed in solution B, centrifuged at 10,000g for 1 min, and resuspended in 400 μl Laemmli sample buffer. The P2 fraction was sonicated four times for 30 s each time using a Bioruptor (Bio-Rad).

**Chromatin immunoprecipitation followed by western blotting.** Approximately $4 \times 10^7$ cells transfected with HA-DDB2 were exposed to ultraviolet 40 J m⁻². Following crosslinking by formaldehyde and neutralization with 1 M glycine, cells were collected and resuspended in 1 ml of lysis buffer1 (50 mM HEPES (pH 7.5), 140 mM NaCl, 1 mM EDTA, 10% Glycerol, 0.5% NP-40, 0.25% Triton X-100 and protease inhibitor), centrifuged, and supernatant was discarded. Resuspend cells in 1 ml of lysis buffer2 (10 mM Tris-HCl (pH 8.0), 200 mM NaCl, 1 mM EDTA, 0.5 mM EGTA and protease inhibitor), centrifuge, discard supernatant and then resuspend pellet in shearing buffer (1% SDS, 10 mM EDTA, 50 mM Tris-HCl (pH 8.0) and protease inhibitor). Sonicate sample and dilute 1:10 with ChIP dilution buffer (50 mM Tris-HCl (pH 8.0), 167 mM NaCl, 1.1% Triton X-100, 0.11% Na-Deoxycholate and protease inhibitor). Split each sample into two groups to incubate with 4 μg of either anti-HA antibody (Santa Cruz) or mouse normal IgG as a control (Santa Cruz) overnight at 4 °C. Chromatin, incubated with the respective antibodies, was purified by precipitating for another 1 h at 4 °C with Dynabeads-Protein-G (Invitrogen). Precipitated chromatin was washed twice with wash buffer 1 (50 mM Tris-HCl (pH 8.0), 0.1% SDS, 0.1% Na-Deoxycholate, 1% Triton X-100, 150 mM NaCl, 1 mM EDTA, 0.5 mM EGTA), once with wash buffer 2 (50 mM Tris-HCl (pH 8.0), 0.1% SDS, 0.1% Na-Deoxycholate, 1% Triton X-100, 500 mM NaCl, 1 mM EDTA, 0.5 mM EGTA), once with wash buffer 3 (50 mM Tris-HCl (pH 8.0), 250 mM LiCl, 0.5% Na-Deoxycholate,

0.5% NP-40, 1 mM EDTA, 0.5 mM EGTA), and twice with wash buffer 4 (50 mM Tris-HCl (pH 8.0), 10 mM EDTA, 5 mM EGTA) before de-crosslinking. Cross-linked chromatin was reversed by incubation with ChIP elution buffer (10 mM Tris-HCl (pH 8.0), 300 mM NaCl, 5 mM EDTA, 0.5% SDS) for 4 h at 65 °C. Immunoprecipitated samples were subjected to western blotting according to the standard methods.

**Statistical analysis.** Statistical analysis was performed by Student's *t*-test.

**Uncropped scans.** Uncropped scans of immunoblots were supplied in Supplementary Fig. 9.

**Data availability.** The authors declare that all relevant data are available within the article and its Supplementary Information Files from the corresponding author upon reasonable request.

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

## Acknowledgements

We thank Dr Galande for critical reading of the manuscript, Ms Kinpara for technical assistance, Dr Yasui for providing expression vectors of chromatin remodellers, Dr Sugasawa for providing expression vector of GFP-XPA, Dr Shibata and Dr Shiotani for helpful discussion, Dr Fujiyama for his kind cooperation and members of Advanced Research Facilities and Services for assistance using TCS SP8 and In Cell Analyzer 2200. This work was supported in part by grants from the Ministry of Education, Culture, Sports, Science and Technology of Japan (to H.N. and M.K.).

## Author contributions

H.N. and R.M. generated constructs and stable cell lines. H.N. and R.M. performed micro-irradiation. R.H. and C.U. performed live-cell imaging experiments. Y.N. performed RRS assay, and A.M. performed clonogenic assay. K.N., S.S., T.O. and K.K. performed the experiments and analysed the data. S.M., H.N., A.U., T.O. and M.K. edited the manuscript. H.N. supervised the project and wrote the paper.

## Additional information

**Competing interests:** The authors declare no competing financial interests.

**DOI: 10.1038/ncomms16214**      **OPEN**

# Author Correction: Phosphorylated HBO1 at UV irradiated sites is essential for nucleotide excision repair

Hiroyuki Niida, Ryoichi Matsunuma, Ryo Horiguchi, Chiharu Uchida, Yuka Nakazawa, Akira Motegi, Koji Nishimoto, Satoshi Sakai, Tatsuya Ohhata, Kyoko Kitagawa, Shinichi Moriwaki, Hideo Nishitani, Ayako Ui, Tomoo Ogi & Masatoshi Kitagawa

*Nature Communications* 8:16102 doi: 10.1038/ncomms16102 (2017); Published online 18 Jul 2017; Updated 20 Apr 2018

The original version of this Article contained an error in Fig. 1b, in which the cells were incorrectly labelled as 'HeLa', rather than the correct 'HEK293'

Also, the following was omitted from the end of the legend of Fig. 1b: 'Scanned images were subjected to linear adjustments of light–dark and contrast by Microsoft Power Point.'

These have been corrected in both the PDF and HTML versions of the Article.

The original version of the Supplementary Information associated with this Article contained errors in Supplementary Figs 3d, 5c, 7 and 9.

In Supplementary Fig. 3d, the shHBO1 image labelled '600' inadvertently repeated those labelled '450'.

In Supplementary Fig. 5c, the images labelled '98BR', '3KA' and '7HM' were previously incorrectly showing 7HM, NMKW and 48BR cell lines, respectively. The correct version of Supplementary Fig. 5c appears below as Fig. 1:

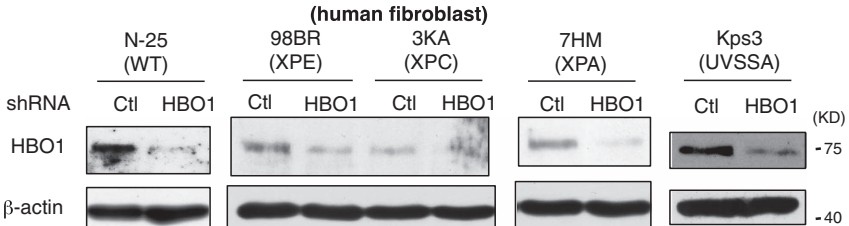

**Figure 1**

which replaces the previous incorrect version, which appears below as Fig. 2:

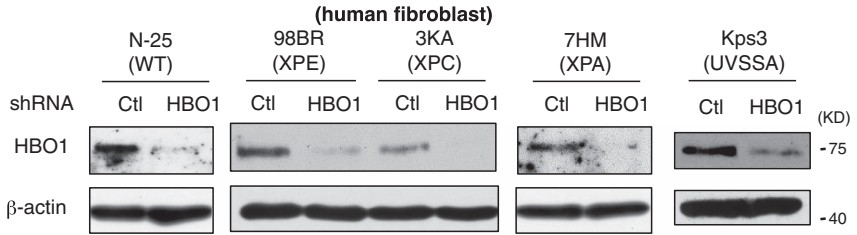

**Figure 2**

In Supplementary Fig. 7, the image displayed the results of one experiment instead of three independent experiments as indicated in the legend. The correct version of Supplementary Fig. 7 appears below as Fig. 3:

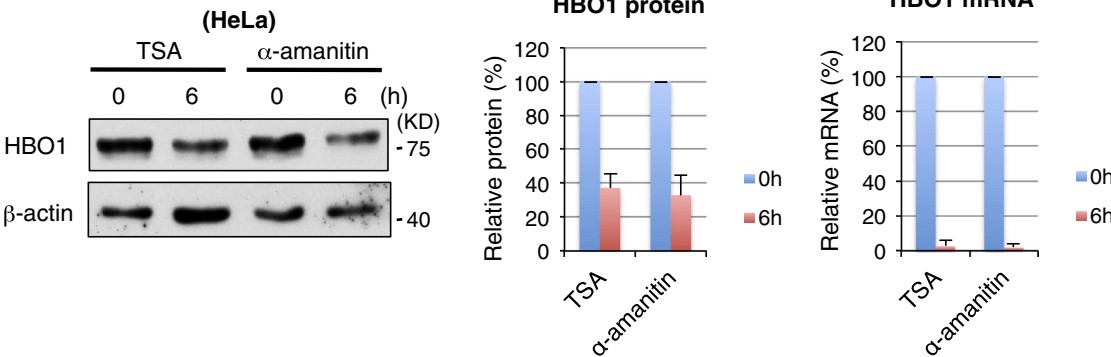

**Figure 3**

which replaces the previous incorrect version, which appears below as Fig. 4:

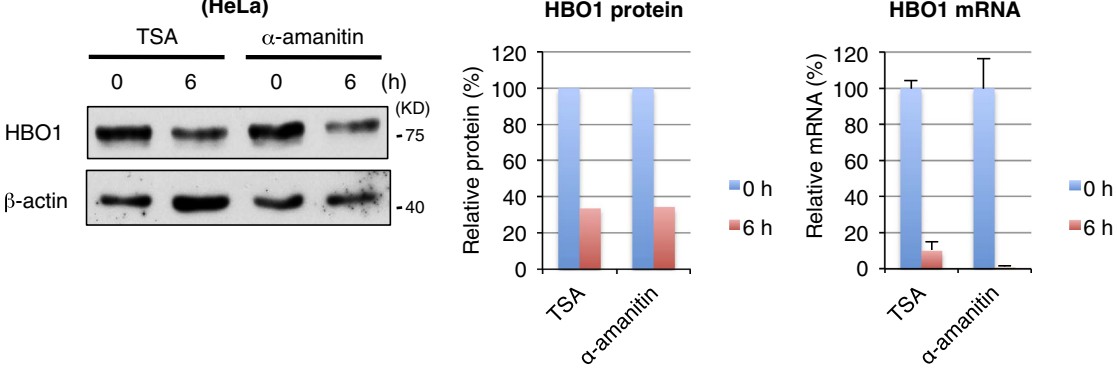

**Figure 4**

In Supplementary Fig. 9, the uncropped images for Fig. 1b below the labels 'pS50/53-' and 'pS50/53 UV' were swapped, as were those below the labels 'HBO1 –' and 'HBO1 UV'. Also, the blue arrow indicating the HBO1 band was missing from the image labelled 'HBO1 –'.

In Supplementary Fig. 9, in the uncropped images for Fig. 2b, the bottom-left image labelled 'HBO1' previously incorrectly presented with an unrelated uncropped blot and the bottom-right image for beta actin incorrectly presented from a different experiment. The correct version of the uncropped images for Fig. 2b given in Supplementary Fig. 9 appears below as Fig. 5:

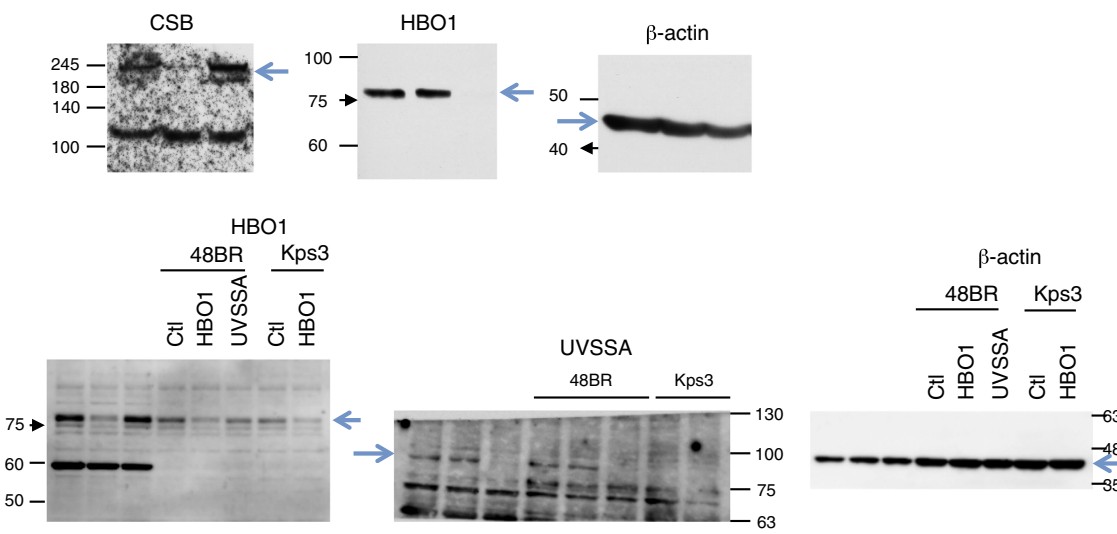

**Figure 5**

which replaces the previous incorrect version, which appears below as Fig. 6:

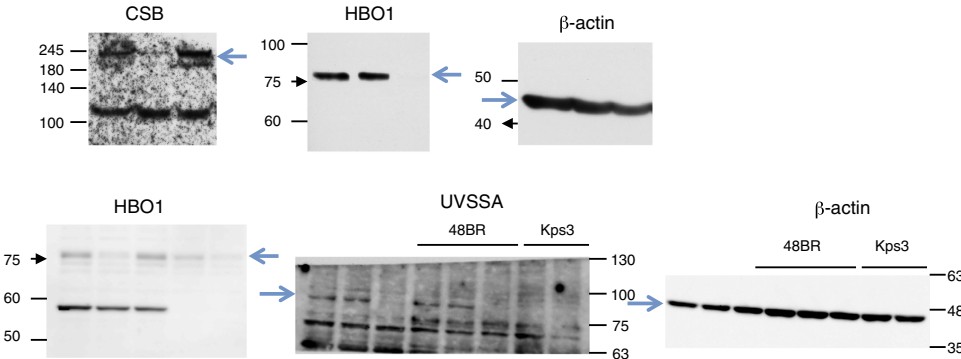

**Figure 6**

In Supplementary Fig. 9, in the uncropped images for Supplementary Fig. 5c, the third image from the left under the label 'HBO1' was previously incorrectly showing 48BR cell lines, rather than the intended 7HM cell lines. The correct version of the uncropped images for Supplementary Fig. 5c given in Supplementary Fig. 9 appears below as Fig. 7:

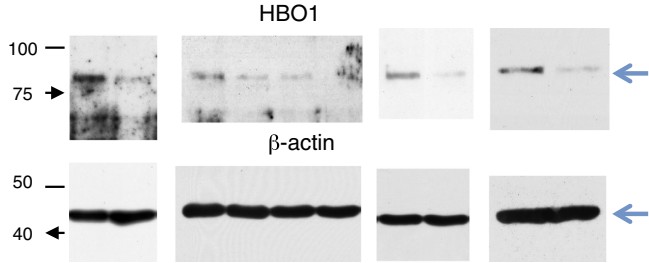

**Figure 7**

which replaces the previous incorrect version, which appears below as Fig. 8:

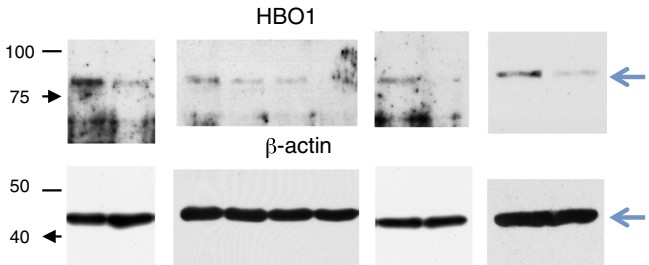

**Figure 8**

The HTML has been updated to include a corrected version of the Supplementary Information.

