## [Peer Review File · Nature Communications]

Reviewers' comments:

Reviewer #1 (Remarks to the Author):

Previously, the authors reported ATM/ATR-dependent S50/53 phosphorylation and CRL4-DDB2-dependent degradation of histone acetyltransferase HBO1 following UV irradiation. Here, the authors report that phosphorylated HBO1 localizes to UV-damage site in a DDB2 dependent manner, which facilitates the subsequent recruitment of XPC. Knockdown of HBO1 furthermore leads to DNA repair defects and increased UV survival. The authors observe HBO1-dependent increase in H3 and H4 acetylation levels after UV and HBO1-dependent recruitment of SNF2H-ACF1 chromatin remodeling complex, and speculate that HBO1-dependent chromatin remodeling further facilitates XPC accumulation at site of damage.

Throughout the manuscript the authors mention that phosphorylated HBO1 is recruited/accumulated/localized to sites of DNA damage (on page 14 even that 'HBO1 accumulation'). However, IF and chromatin fractionation experiments in Fig 1a and b suggest that HBO1 is already present on the DNA/chromatin (in contrast to the chromatin recruitment of DDB2 and XPC in Fig 1B) and is phosphorylated at the site of damage. The authors present no evidence that HBO1 is first phosphorylated and then recruited to damage. This should be corrected and/or discussed. Furthermore, the experiment in Fig 1B would be strengthened if the authors could show that HBO1 in the nucleoplasm is not phosphorylated upon UV damage (where is the non-chromatin fraction?).

In general is the image quality of IF very poor. Therefore, co-localization is difficult to assess from the provided images. Also in Fig 1d (and Fig 1e), no positive control damage staining is included and shown, making it impossible to quantitatively judge whether indeed less pS0/53 signal is present at local damage after shDDB2 or whether the intense pS0/53 signal in the shXPC cell really represents a damaged site. In these experiments, control damage staining (DDB2 or CPD) should be included and co-localization should be quantified.

To perform proper quantification of the Unscheduled DNA synthesis assays (Fig 2a), the authors should exclude EdU signals from S-phase cells from their measurements. According to the flow cytometry image in figure 2a, a considerable amount of cells (actual cell numbers are not indicated on the Y-axis) has gone through S phase during the 3 hrs EdU labeling period. How have the authors excluded EdU signals from these cells? Based on the provided poor quality images of the cells in the top panel, this seems rather difficult to do properly. Also, in their previous paper (Matsunuma, MCB 2016), the authors reported that cells with mutant HBO1 show cell cycle regulation defects after UV damage. Therefore, considering potential differences in amount of S phase cells between shCtl and shHBO1, the authors should measure only EdU incorporation in non-S phase cells.

Page 11. To show the importance of HBO1 phosphorylation and HAT activity for XPC recruitment, the authors express Myc-tagged HBO1 wild type and mutants in HBO1 siRNA depleted cells. It is not clearly indicated whether these Myc-tagged HBO1 constructs are resistant to the siRNA treatment, the authors should show expression of the ectopically expressed constructs. Because the observed differences in XPC recruitment are not very big, expression levels of Myc-tagged wild type and mutant HBO1 proteins should be determined, in comparison to endogenous HBO1, to rule out effects of residual endogenous HBO1 and/or differences in HBO1 (wt or mutant) expression levels and thus to be able to draw definite conclusions from these experiments. The same holds true for experiments shown in fig 2e and 4b and d. It would be even better and much more convincing to generate cell lines that stably express equal amounts of HBO1 WT and mutant proteins. Also, the effect of HBO1 overexpression on XPC recruitment should be determined. In addition, to confirm (partial) involvement of HBO1 phosphorylation in regulation of XPC recruitment, the authors should test whether ATM/ATR inhibition indeed leads to diminished XPC recruitment.

The authors use 405 nm laser irradiation (with or without Hoechst 33342) to study NER-related recruitment of XPC and SNF2H. Although XPC is known to be recruited to 405 nm laser induced DNA damage, depending on the precise methodology such UV-A laser treatment produces predominantly oxidative base damage, single strand breaks and no significant amounts of CPD

photo-lesions, and does not activate NER (even though XPC is recruited to these UV-A-induced lesions, see Menoni et al, JCB, 2012). Therefore, the authors should show activation of NER by this method, either by showing that downstream NER proteins are also recruited or that CPDs are indeed produced (with and without Hoechst 33342). SNF2H and ACF1 were shown before to be recruited to UV photolesions, oxidative damage and double strand breaks, using 405 nm and other types of lasers, and to be involved in repair of all these types of lesions (e.g. Lan et al Mol Cell 2010; Erdel and Rippe, Nucleus, 2011; Sanchez-Molina et al, NAR, 2011; Aydin et al, NAR, 2014; Kubota et al, DNA repair, 2016). Therefore, the authors should understand what type of damage is induced with their laser set-up. They should also properly cite these previous papers describing the different DNA repair roles for SNF2H-ACF1.

Page 11, figs 2d,e. The results presented in fig 2d, e are in direct contrast to results presented in fig 1d and e. In Fig 1 (and page 9) the authors describe that XPC (and TFIIH) recruitment 30 min after DNA damage induction depends on DDB2 and HBO1, but in fig 2 (and page 11) no XPC recruitment defects are observed 20 min after DNA damage induction. This casts doubt on validity of results presented in these figures and should be clarified. Also, the fact that in IF ~10 % more XPC is bound 60 minutes after damage induction at CPD sites after depletion of DDB2 or HBO1 does not have to signify that XPC fails to dissociate. Repair could simply be less efficient and therefore more XPC is found bound at later time point. If the authors wish to measure dissociation of XPC, they should properly measure its K_{on} and K_{off} .

Fig 3a. In the panel with the ChIP blots, also HA-DDB2 levels (after IP) should be shown; particularly since after UV more DDB2 is bound to chromatin. In Fig 3b, in the input H3AcK14 levels (and possibly even H4ac) are lower after depletion of HBO1, suggesting H3 acetylation levels could be lower due to HBO1 depletion even in the absence of DNA damage. Therefore, to judge whether the dependency on HBO1 is UV-specific, authors should perform the same experiment without UV-irradiation.

Lines 205-206 and fig 4d. If the authors wish to show that WT HBO1 protein complements UV survival of HBO1-depleted cells, than they should in the same experiment also measure (and show) UV survival of HBO1 depleted cells (without complementation). This is especially important because DDB2- and shHBO1-depleted cells in fig 4c seem equally UV sensitive as shCtl and WT cells from Fig 4d.

lines 241-246. The authors use γ H2AX staining to prove that their UV irradiation does not induce double strands breaks. Although it is not expected that 50 J/m² of UV-C induces this type of damage, the use of γ H2AX staining will not be informative for this purpose. H2AX is not only phosphorylated after double strand break induction, but also at single strands breaks and thus also during NER. This was shown by several labs before (e.g. O'Driscoll et al, Nat Genet, 2003; Hanasoge and Ljungman, Carcinogenesis, 2007; Martelijn et al, JCB, 2009). Therefore it is rather peculiar that the authors do not observe any γ H2AX staining, as if NER is not activated. Thus, the authors should perform control experiments to test whether their γ H2AX staining protocol is functional. If the authors want to exclude ACF1 localization to UV-generated double strands breaks, they should test ACF1 recruitment in non-replicating cells.

Fig 4d and 5c. Particularly in Fig 5, extreme high doses of UV are used to determine clonogenic UV-survival, which does not match to the plethora of literature on UV-survival, as well as the shape of the survival curves.

Figs 4a, 4b, 5b: The authors should also comment on why at the first time point after UV-irradiation (likely 1 h, not clear from the description/figure legends) approximately 2,5 times more CPD are detected than directly after UV (0 h time point).

Is this relative amount of CPDs of 0.4 background level of CPD detection or really at 0 h after UV?

Fig 6b, lines 254-255. As control for the HBO1-dependent GFP-SNF2H recruitment, the authors should include DDB2 depletion to show specificity of SNF2H recruitment, as HBO1 is claimed to be DDB2 dependent and thus specific for GG-NER, this is particularly important as it was previously reported that SNF2H is implicated in TC-NER (Aydin, 2014).

Fig 6c, lines 259-260. The reciprocal IPs with GFP-SNF2H do not convincingly show that Myc-HBO1

is co-immunoprecipitated but HA-DDB2 not. There is hardly any Myc-HBO1 in the IP and in the input there is huge overexpression of Myc-HBO1 (compared to HA-DDB2). Fig 6e and f, lines 276-284. The authors should show and compare efficiency of SNF2H depletion in all different cell lines, before being able to draw any conclusion on additional defects of SNF2H and HBO1 depletion. Also, the authors have not shown any chromatin remodeling activity of SNF2H-ACF1 and can therefore not conclude anything on 'cooperative chromatin remodeling activity'.

Importantly, also the specificity of shRNA-mediated depletion (this holds for all presented siRNA and shRNA experiments) should be shown, excluding possible off target effects of siRNA and shRNA, preferably by using different shRNA and siRNA.

lines 321-322. This is not true. Molecular mechanisms of ACF1-SNF2H complex recruitment to UV damage has been investigated before, including involvement of histone acetylation. See Aydin et al, NAR, 2014. The authors should discuss similarities and discrepancies with findings of this paper.

Supplementary figure 6 should be included in the results section.

It should be explained why different doses of UV are used in different IF experiments. Now, in Fig1a, 2d and 5a 12 J/m² is used (which seems rather low) and in fig. 1c-e is 50 J/m²

Minor comments

line 54, "deficient GG-NER pathway results in predisposition to the autosomal recessive skin cancer, xeroderma pigmentosum". Please rephrase as xeroderma pigmentosum is not similar to recessive skin cancer. Skin cancer predisposition is one of the symptoms of XP.

line 55-56, XPC is GG-NER initiator in eukaryotic cells, not only in mammalian cells. The authors state that XPC binding is to 'modify and relax the chromatin structure'. It is not clear on what finding this statement is based on? Indeed XPC is the initiator of GG-NER, though in none of the provided references (1 and 2) no information is provided on chromatin modification/relaxation by XPC.

Given that the authors do not show that HBO1 itself is recruited (but phosphorylated at sites of UV damage), the title is misleading.

line 60-68 As RBX1 is also part of the E3-ligase complex, it would be more appropriate and consistent (also with the authors previous paper) to refer to this complex as CRL4-DDB2 instead of CUL4/DDB2.

lines 79-87 ALC1 is not the catalytic subunit of CHD even though its alternative name is CHD1L (=CHD1-like). CHD complexes have thus far not been found to be implicated in nucleotide excision repair. INO80 has been implicated (Jiang et al, PNAS 2010) and should be mentioned here as well. Furthermore, BRG1 was also implicated in mammalian NER by Zhang et al, Cell Cycle, 2009.

line 119. The authors state they have previously shown degradation of HBO1 starting from 2 h after UV. However, this is not detected in the experiment shown in Fig 1b. Do the authors still observe this degradation?

Line 188-189. The authors cannot draw the conclusion that XPC recruitment is facilitated by acetylation of H3 and H4 by HBO1, as they have not shown this directly. They only show a HBO1-dependent increase of H3/H4 acetylation levels after UV. Also, on page 11 the authors themselves already suggest that an additional function of HBO1 other than HAT activity may facilitate XPC recruitment, which contrast this conclusion drawn in lines 188-189.

Line 234, considering that the authors report involvement of SNF2H and ACF1 in NER, they should cite here that SNF2H and ACF1 were already found to be involved in transcription-coupled NER before (Aydin et al, NAR, 2014). Also, INO80 involvement should be cited (Jiang et al, PNAS 2010)

Line 236 Please clarify what is meant with 'Our examination after local-UV irradiation'. Have the authors tested recruitment (to what type of damage?) of multiple ATP-dependent chromatin remodelers and found only ACF1 and SNF2H to be recruited?

Fig 1d Images are not equally sized

line 265 'physically interacts with SNF2H and recruits the ACF1-SNF2H complex to CPD sites'. The authors only show co-IP, although they do not provide evidence for direct physical interaction, can be through chromatin or other complex partners).

lines 296-299. HBO1-depleted cell could have a defect in overall GG-NER, i.e. also in the removal of 6-4PPs, which was not tested. The authors should mention this possibility.

lines 302-307 How do the authors envision that ATM and/or ATR are activated after UV damage to phosphorylate HBO1? Is this happens during NER (and not because of DNA break induction in replicating cells), standard models predict that damage first needs to be processed by NER before ATR can be activated. How could ATM be activated? The authors should discuss this difficult to understand issue.

Reviewer #2 (Remarks to the Author):

The manuscript by Hiroyuki Niida is a continuation of a previous work (it is cited within this work as well) where the authors have shown that HBO1 is predominantly phosphorylated by ATR and interacts with DDB2 upon exposure to UV irradiation. Here, the authors address the functional role of HBO1 in the repair of UV-induced DNA lesions. They first show that HBO1 is phosphorylated by ATR and that it requires DDB2 to recruit to UV-induced DNA lesions. They next show that HBO1 acetylates histone H3 Lys 14 and histone H4 aimed at facilitating the recruitment of XPC at sites of DNA damage (i.e. cyclobutane pyrimidine dimers) upon UV irradiation. In line, shHBO1 cells show delayed CPD repair and are sensitive to UV. The authors conclude that HBO1 plays a role during the early steps of the GG-NER subpathway in NER.

The data shown in this work are significant, of interest to the field, they are clearly presented and for the most part they seem to support the conclusions drawn in this work. The manuscript could benefit from further editing.

A major concern is that the authors do not provide adequate evidence for the specificity of HBO1 to UV-induced DNA CPDs as compared to i. other types of UV-induced DNA damages (6-4PPs), ii. to other types of DNA damages in general (DNA ICLs, DNA breaks, etc.) or even to iii. the TC-NER sub-pathway of NER. Supporting the specificity of their findings will further enhance the significance and scope of the work.

1. Do the authors see a similar retention of HBO1 to chromatin when cells are exposed to the e.g. DNA cross-linker mitomycin (for DNA ICLs) or upon IR? It would be important to address the specificity of HBO1 to UV-induced DNA damage.
2. The authors provide evidence that "Functional HBO1 is required for efficient GG-NER". The authors need to provide a clear justification and preferably experimental evidence on why they think the role of HBO1 in TC-NER is not relevant or that it should be excluded. Next to UDS and RNA synthesis recovery assay would be useful.
3. The blots on Ac-H4 are not clear. If possible some clearer, higher quality images should be provided.
4. Are shHBO1 cells sensitive to other type of DNA damages (besides UV)?
5. Could the authors use separate inhibitors to distinguish whether HBO1 role requires ATM or ATR specifically? Otherwise they should provide a justification why when ATR was shown to predominantly phosphorylate HBO1, the authors use an ATR/ATM inhibitor.
6. Do CSB or CSA cells show (or not) additional UV sensitivity following depletion of HBO1?
7. The authors use γ H2AX as an indicator of DSB sites. As such they assume and subsequently show no γ H2AX foci in UV-treated cells. This is somewhat surprising as γ H2AX foci are known to be detected even in 10% of non-irradiated cells and do increase remarkably upon UV irradiation (several papers exist). The authors need to re-consider where they stand by their original observation.

Reviewer #1 (Remarks to the Author):

Previously, the authors reported ATM/ATR-dependent S50/53 phosphorylation and CRL4-DDB2-dependent degradation of histone acetyltransferase HBO1 following UV irradiation. Here, the authors report that phosphorylated HBO1 localizes to UV-damage site in a DDB2 dependent manner, which facilitates the subsequent recruitment of XPC. Knockdown of HBO1 furthermore leads to DNA repair defects and increased UV survival. The authors observe HBO1-dependent increase in H3 and H4 acetylation levels after UV and HBO1-dependent recruitment of SNF2H-ACF1 chromatin remodeling complex, and speculate that HBO1-dependent chromatin remodeling further facilitates XPC accumulation at site of damage.

1. Throughout the manuscript the authors mention that phosphorylated HBO1 is recruited/accumulated/localized to sites of DNA damage (on page 14 even that 'HBO1 accumulation'). However, IF and chromatin fractionation experiments in Fig 1a and b suggest that HBO1 is already present on the DNA/chromatin (in contrast to the chromatin recruitment of DDB2 and XPC in Fig 1B) and is phosphorylated at the site of damage. The authors present no evidence that HBO1 is first phosphorylated and then recruited to damage. This should be corrected and/or discussed. Furthermore, the experiment in Fig 1B would be strengthened if the authors could show that HBO1 in the nucleoplasm is not phosphorylated upon UV damage (where is the non-chromatin fraction?).

We understand the reviewer's suggestions. We have changed the wording of "HBO1 is recruited/accumulated/localized to sites of DNA damage" in the manuscript to "HBO1 is maintained at DNA damage sites" (line 1, line 41, line 111, line 139, line 163, line 168, line 293, line 307 and line 870). We also included discussion that HBO1 does not accumulate at damage sites but rather that HBO1 on chromatin is phosphorylated at damage sites in the discussion section as follows:

"...in contrast to ACF1-SNF2H protein accumulated at DNA damage sites, HBO1 protein amounts did not significantly change at chromatin sites in response to DNA damage. As HBO1 protein is already abundant on the chromatin, we speculate that HBO1 that is already present on chromatin is phosphorylated by ATR in response to UV-induced DNA damage." (line 477-481)

Furthermore, we showed little phosphorylation of HBO1 in the soluble fraction including nucleoplasm in Fig. 1b.

2. In general is the image quality of IF very poor. Therefore, co-localization is difficult to assess from the provided images. Also in Fig 1d (and Fig 1e), no positive control damage staining is included and shown, making it impossible to quantitatively judge whether indeed less pS0/53 signal is present at local damage after shDDB2 or whether the intense pS0/53 signal in the shXPC cell really represents a damaged site. In these experiments, control damage staining (DDB2 or CPD) should be included and co-localization should be quantified.

We have followed these suggestions. We performed double immunostaining of DDB2 and XPC in shControl, shDDB2, shHBO1 and shXPC cells (Fig. 1d upper panel) and quantitated the relative fluorescence intensities of XPC toward DDB2 in shCtl and shHBO1 cells (Fig. 1d bottom left graph). We also performed double immunostaining of control DDB2 and HBO1 pS50/53 in shControl and shXPC cells (Fig. 1d bottom panel). We quantified the relative fluorescence of HBO1 pS50/53 toward DDB2 (Fig. 1d bottom right graph). We further performed double immunostaining of control DDB2 and TFIIH in shControl and shHBO1 cells and quantified the relative fluorescence of TFIIH toward DDB2 (Supplementary Fig. 1e). Furthermore, to show the defect of XPC accumulation at UV-induced damage sites in HBO1 depleted cells, we also quantified the relative fluorescence of XPC toward CPD after 20 min in shControl, shDDB2 and shHBO1 cells (Fig. 3f). We compared the relative fluorescence of XPC toward CPD between stable depleted endogenous HBO1 cells with shHBO1-resistant Myc-WT, EQ or SA HBO1 cells (Fig. 3h).

Collectively, these results suggested that HBO1 facilitated accumulation of XPC at sites of UV-induced damage and that HBO1 could stay at damage sites without XPC.

3. To perform proper quantification of the Unscheduled DNA synthesis assays (Fig 2a), the authors should exclude EdU signals from S-phase cells from their measurements. According to the flow cytometry image in figure 2a, a considerable amount of cells (actual cell numbers are not indicated on the Y-axis) has gone through S phase during the 3 hrs EdU labeling period. How have the authors excluded EdU signals from these cells? Based on the provided poor quality images of the cells in the top panel, this seems rather difficult to do properly. Also, in their previous paper (Matsunuma, MCB 2016), the authors reported that cells with mutant HBO1 show cell cycle regulation defects after UV damage. Therefore, considering potential

differences in amount of S phase cells between shCtl and shHBO1, the authors should measure only EdU incorporation in non-S phase cells.

We agree with this reviewer's comment. We thus selected only G1 cells from synchronized cells by quantification of DNA contents using the IN CELL Analyzer 2200 (GE Healthcare). Populations of analysed cells are shown in Supplementary Fig. 2. The results of unscheduled DNA synthesis in shControl, shDDB2 and shHBO1 cells in G1 phase are now shown in Fig. 2a.

4. Page 11. To show the importance of HBO1 phosphorylation and HAT activity for XPC recruitment, the authors express Myc-tagged HBO1 wild type and mutants in HBO1 siRNA depleted cells. It is not clearly indicated whether these Myc-tagged HBO1 constructs are resistant to the siRNA treatment, the authors should show expression of the ectopically expressed constructs. Because the observed differences in XPC recruitment are not very big, expression levels of Myc-tagged wild type and mutant HBO1 proteins should be determined, in comparison to endogenous HBO1, to rule out effects of residual endogenous HBO1 and/or differences in HBO1 (wt or mutant) expression levels and thus to be able to draw definite conclusions from these experiments. The same holds true for experiments shown in fig 2e and 4b and d. It would be even better and much more convincing to generate cell lines that stably express equal amounts of HBO1 WT and mutant proteins. Also, the effect of HBO1 overexpression on XPC recruitment should be determined. In addition, to confirm (partial) involvement of HBO1 phosphorylation in regulation of XPC recruitment, the authors should test whether ATM/ATR inhibition indeed leads to diminished XPC recruitment.

We showed expression levels of Myc-HBO1 WT, EQ and SA together with endogenous HBO1 in shControl and shHBO1 cells (Supplementary Fig. 3d). We also showed that endogenous HBO1 protein levels in shHBO1 cells were less than 20% of shControl cells in a previous report (Ref. 15, Matsunuma et al, Mol. Cell Biol. 2015, Supplementary Fig 3c). The western blot showed that expressions of Myc-HBO1 WT, WQ and SA were similar and comparable to endogenous HBO1 in shControl cells. We have now included text stating that Myc-tagged HBO1 is resistant to HBO1 shRNA in the text (line 215, line 239, line 276). We also included this information in the figure legend of Supplementary Fig. 3d.

To evaluate HBO1-WT, -EQ and -SA, endogenous HBO1 was first stably depleted by shHBO1 and then myc-HBO1-WT, -WQ and -SA were stably transfected. We included discussion of these experiments as follows:

“we monitored the accumulation of XPC-EGFP in cells stably expressing shHBO1-resistant Myc-tagged HBO1-WT (wild-type HBO1) or the mutants HBO1-EQ (Glu 508 substituted to Gln, catalytically dead) or HBO1-SA (Ser50 and 53 substituted to Ala) with concurrent shRNA depletion of endogenous HBO1 (Fig. 3b, Supplementary Fig. 3d and e).” (line 214-218)

“Consistent with the ability of HBO1 to facilitate XPC loading, removal of CPDs was significantly delayed in stable HBO1-depleted cells expressing shHBO1-resistant HBO1-EQ or -SA mutants (Fig. 5b).” (line 274-276)

We thus concluded that differences of XPC recruitment between Myc-HBO1 WT and mutants were due to defects of HAT activity and lack of phosphorylation of Ser50 and 53 in HBO1.

We confirmed that ATM/ATR inhibition led to diminished XPC recruitment by live cell imaging of XPC-EGFP accumulation (Fig. 3c) and micropore UV exposure experiments (Fig. 3d).

5. The authors use 405 nm laser irradiation (with or without Hoechst 33342) to study NER-related recruitment of XPC and SNF2H. Although XPC is known to be recruited to 405 nm laser induced DNA damage, depending on the precise methodology such UV-A laser treatment produces predominantly oxidative base damage, single strand breaks and no significant amounts of CPD photo-lesions, and does not activate NER (even though XPC is recruited to these UV-A-induced lesions, see Menoni et al, JCB, 2012). Therefore, the authors should show activation of NER by this method, either by showing that downstream NER proteins are also recruited or that CPDs are indeed produced (with and without Hoechst 33342). SNF2H and ACF1 were shown before to be recruited to UV photolesions, oxidative damage and double strand breaks, using 405 nm and other types of lasers, and to be involved in repair of all these types of lesions (e.g. Lan et al Mol Cell 2010; Erdel and Rippe, Nucleus, 2011; Sanchez-Molina et al, NAR, 2011; Aydin et al, NAR, 2014; Kubota et al, DNA repair, 2016). Therefore, the authors should understand what type of damage is induced with their laser set-up.

They should also properly cite these previous papers describing the different DNA repair roles for SNF2H-ACF1.

We appreciate this helpful suggestion. We examined whether 405 nm laser irradiation with or without Hoechst 33342 generated CPD regions and recruited GFP-XPA at the lesions. We found that 405 nm laser irradiation with Hoechst 33342 generated significant CPD (Supplementary Fig. 3a left panel) and GFP-XPA was recruited at the irradiated regions (Supplementary Fig. 3a right panel). However, we could not detect generation of CPD nor GFP-XPA recruitment after 405 nm laser irradiation without Hoechst 33342. We thought that GFP-SNF2H accumulation in Fig. 6b in the previous version of the manuscript was not responsive to CPD. Therefore, we would like to replace the previous result that was performed without Hoechst 33342 with the new experiment performed with Hoechst 33342 (Fig. 7b). We have explained the new result in the text as follows:

“We further examined SNF2H accumulation by monitoring GFP-SNF2H in living cells (Fig. 7b). Although 405 nm laser irradiation with Hoechst 33342 in the culture medium induces not only CPDs but also DSBs in DNA³⁶, we found reduced accumulation of GFP-SNF2H in shDDB2 and shHBO1 cells (Fig. 7b).” (line 354-357)

We understood the kinds of DNA damage generated by 405 nm laser irradiation with Hoechst 33342 and the role of SNF2H-ACF1 in different repair pathways from previous papers. We have cited these papers in our manuscript and have described this discussion in the text (line 201-204, line 332-333, line 336-337, line 341-346).

6. Page 11, figs 2d,e. The results presented in fig 2d, e are in direct contrast to results presented in fig 1d and e. In Fig 1 (and page 9) the authors describe that XPC (and TFIIH) recruitment 30 min after DNA damage induction depends on DDB2 and HBO1, but in fig 2 (and page 11) no XPC recruitment defects are observed 20 min after DNA damage induction. This casts doubt on validity of results presented in these figures and should be clarified. Also, the fact that in IF ~10 % more XPC is bound 60 minutes after damage induction at CPD sites after depletion of DDB2 or HBO1 does not have to signify that XPC fails to dissociate. Repair could simply be less efficient and therefore more XPC is found bound at later time point. If the authors wish to measure dissociation of XPC, they should properly measure its K_{on} and K_{off} .

Based on the reviewer's suggestion, we have clarified the results of the previous Fig. 2d and e (new Fig. 3e and g) indicating co-localization of CPD and XPC. These figures did not mean relative accumulation of XPC at damage sites. To clarify these points, we have added the relative accumulation of XPC to damage sites in these experiments in Fig. 3f and h. We have described these points in the text as follows:

“We therefore measured the rate of co-localization and relative quantity of XPC at CPD sites at a later time point (Fig. 3e and f, respectively). Twenty minutes after 12 J/m² UV irradiation, XPC accumulated in about 70% of the CPD-positive areas in HBO1-depleted cells. The accumulation level of XPC in CPD-positive areas in HBO1-depleted cells was almost the same as in mock-depleted or DDB2-depleted cells (Fig. 3e). At this time point, the relative quantity of XPC at CPD positive areas in shHBO1 and shDDB2 cells was 67% and 62% of shCtl cells, respectively (Fig. 3f). In cells depleted of DDB2 or HBO1, XPC co-localization with CPD was still abundant at DNA damage sites 60 min after UV irradiation (Fig. 3e). The extended localization of XPC because of the defect of rapid recruitment in HBO1-depleted cells was rectified by stable expression of shHBO1-resistant HBO1-WT, but not HBO1-EQ or -SA mutants (Fig. 3g). In this rescue experiment, the relative quantity of XPC at CPD positive areas at 20 min after UV was only complemented by stable expression of HBO1-WT (Fig. 3h).” (line 228- 242)

We speculated that longer localization of XPC at damage sites were results of the defect of rapid recruitment in HBO1-depleted cells. We described this point in the text as follows:

“In cells depleted of DDB2 or HBO1, XPC co-localization with CPD was still abundant at DNA damage sites 60 min after UV irradiation (Fig. 3e). The extended localization of XPC because of the defect of rapid recruitment in HBO1-depleted cells...” (line 235- 238)

7. Fig 3a. In the panel with the ChIP blots, also HA-DDB2 levels (after IP) should be shown; particularly since after UV more DDB2 is bound to chromatin. In Fig 3b, in the input H3AcK14 levels (and possibly even H4ac) are lower after depletion of HBO1, suggesting H3 acetylation levels could be lower due to HBO1 depletion even in the absence of DNA damage. Therefore, to judge whether the dependency on HBO1 is UV-specific, authors should perform the same experiment without UV-irradiation.

According to the reviewer's suggestion, we showed HA-DDB2 after IP in Fig. 4a-c. Immunoprecipitated HA-DDB2 showed similar amounts in each lane.

We also understand the reviewer's suggestion about histone H3 AcK14 in shHBO1 cells. To determine that the dependency of acetylation on histone H3AcK14 is UV-specific, we performed ChIP-western blotting with shControl and shHBO1 cells without UV irradiation. The amount of histone H3AcK14 in shHBO1 cells after IP was about 75% of control cells. These results are shown in Fig. 4c. We thus concluded that histone H3AcK14 around DDB2-bound chromatin was increased in an HBO1-dependent manner and have described this in the text as follows,

“The global acetylation level of histone H3 at Lys14 in shHBO1 cells was reduced, and therefore the reduction of histone H3 Lys14 immunoprecipitated by HA-DDB2 might reflect global hypo-acetylation of histone H3 Lys14. To test this possibility, we performed ChIP-western blot with non-irradiated shHBO1 cells (Fig. 4c). Acetylated histone H3 Lys14 immunoprecipitated from shHBO1 cells was about 0.8 fold compared with shCtl cells (Fig. 4c). Based on these results, we concluded that UV-induced acetylation of H4 and H3 Lys14 around sites of UV lesions was HBO1-dependent.” (line 254-261)

8. Lines 205-206 and fig 4d. If the authors wish to show that WT HBO1 protein complements UV survival of HBO1-depleted cells, than they should in the same experiment also measure (and show) UV survival of HBO1 depleted cells (without complementation). This is especially important because DDB2- and shHBO1-depleted cells in fig 4c seem equally UV sensitive as shCtl and WT cells from Fig 4d.

We appreciate this suggestion. We have added the results of the same experiment using only shHBO1 in Fig. 5d (previous Fig. 4d).

9. lines 241-246. The authors use Fig 4d and 5c. Particularly in Fig 5, extreme high doses of UV are used to determine clonogenic UV-survival, which does not match to the plethora of literature on UV-survival, as well as the shape of the survival curves.

We appreciate this helpful observation. We have removed plots of high dose (60 J/m²) UV from Fig. 6c (previous Fig. 5c).

10. Figs 4a, 4b, 5b: The authors should also comment on why at the first time point after UV-irradiation (likely 1 h, not clear from the description/figure legends) approximately 2,5 times more CPD are detected than directly after UV (0 h time point). Is this relative amount of CPDs of 0.4 background level of CPD detection or really at 0 h after UV?

We apologize for the insufficient explanations for these figures. The time point 0 h means no UV irradiation, and therefore 0.4 CPD at 0 h was the background level. We have now described this information **in the figure legends of Fig. 5a, b and Fig. 6b** (previous Fig. 4 a, b and Fig. 5b).

11. Fig 6b, lines 254-255. As control for the HBO1-dependent GFP-SNF2H recruitment, the authors should include DDB2 depletion to show specificity of SNF2H recruitment, as HBO1 is claimed to be DDB2 dependent and thus specific for GG-NER, this is particularly important as it was previously reported that SNF2H is implicated in TC-NER (Aydin, 2014).

We understand this comment and have performed the experiment of GFP-SNF2H recruitment in live cells. Consistent with our observation in shHBO1 cells, SNF2H recruitment was compromised in DDB2-depleted cells. We have added this result in **Fig. 7b**. We further examined the effects of inhibition of transcription and histone de-acetylation, which were used in the previous report, on HBO1 expression. We demonstrated that those treatments strongly inhibited HBO1 expression in **Supplementary Fig. 7**. We have discussed our explanation of the different results from Aydin *et al.* and our study in the Discussion section as follows:

“To understand the relationship between transcription or histone deacetylation-dependent and HBO1-dependent SNF2H recruitment at UV damage sites, we quantified HBO1 protein and mRNA expression after α -amanitin and TSA treatment. We discovered that both treatments strongly repressed HBO1 mRNA expression. As a consequence, HBO1 protein rapidly decreased to about 30% of control. Therefore, defects of SNF2H recruitment by α -amanitin and TSA reagents would partly cause of reduction of HBO1 protein. Regarding the contrasting results of UDS and RRS assays in HBO1- and SNF2H-depleted cells, we speculated that this is the result of HBO1-dependent and -independent SNF2H recruitments. The results of UDS assays in HBO1 depleted cells might indicate that acetylation of histones is more important for GG-NER than chromatin remodelling by ACF1-SNF2H. However, SNF2H would be supplied

for a TC-NER HBO1-independent mechanism. ISWI is thought to bind near the transcription start site⁴⁴ and speculated to facilitate an open chromatin structure for efficient CSB association²⁶. In general, histones of transcriptionally active genes are acetylated⁴⁵, and therefore SNF2H might be able to scan and bind to target nucleosome in the vicinity of lesions without HBO1.” (line 448- 464)

12. Fig 6c, lines 259-260. The reciprocal IPs with GFP-SNF2H do not convincingly show that Myc-HBO1 is co-immunoprecipitated but HA-DDB2 not. There is hardly any Myc-HBO1 in the IP and in the input there is huge overexpression of Myc-HBO1 (compared to HA-DDB2).

In response to the suggestion, we have replaced the previous short-exposed panel to the longer exposed DDB2 panel, which was exposed to X-ray film as long as the HBO1 exposure. The new panel shows a similar overexpression level of HA-DDB2 as Myc-HBO1 (Fig. 7c right panel).

13. Fig 6e and f, lines 276-284. The authors should show and compare efficiency of SNF2H depletion in all different cell lines, before being able to draw any conclusion on additional defects of SNF2H and HBO1 depletion. Also, the authors have not shown any chromatin remodeling activity of SNF2H-ACF1 and can therefore not conclude anything on 'cooperative chromatin remodeling activity'.

In response to the reviewer’s suggestion, we have shown efficiency of SNF2H depletion in all cell lines (Fig. 7e).

We agree with the reviewer’s suggestion about the statement on chromatin remodelling activity in the previous manuscript and have therefore removed the sentence of “cooperative chromatin remodelling activity” in the revised manuscript.

14. Importantly, also the specificity of shRNA-mediated depletion (this holds for all presented siRNA and shRNA experiments) should be shown, excluding possible off target effects of siRNA and shRNA, preferably by using different shRNA and siRNA.

We followed this suggestion. We used siRNAs with different target sequences and performed experiments with these new siRNAs. New shHBO1 (shHBO1-2) and siDDB2 (siDDB2-2) were

used for the experiments of Supplementary Fig. 3c. New siXPC (siXPC-2) was used for immunofluorescence of Supplementary Fig. 1c. New siSNF2H (siSNF2H-2) was used for the experiment of Supplementary Fig. 8b. The offtarget effect of siUVSSA was already evaluated in a previous paper (Ref. 27). We used two siCSB sequences (siCSB and siCSB-2) to perform clonogenic assays in Supplementary Fig. 5b.

15. lines 321-322. This is not true. Molecular mechanisms of ACF1-SNF2H complex recruitment to UV damage has been investigated before, including involvement of histone acetylation. See Aydin et al, NAR, 2014. The authors should discuss similarities and discrepancies with findings of this paper.

We appreciate the helpful comment clarifying this sentence. We have corrected the manuscript and discussed similarities and discrepancies as follows:

“ACF1 and SNF2H have been identified as chromatin remodellers that accumulate at UV-induced damage^{26, 32} and DSB sites³⁴, respectively. SNF2H targeting to UV-C damage sites depends on transcription and histone modification to recruit CSB. These cells showed defects in RRS but UDS was normal²⁶. The authors concluded that SNF2H facilitated TC-NER. ” (line 438-442)

“Here we demonstrated that SNF2H, the ATPase subunit of the ACF1 complex, binds to HBO1 and HBO1-dependently localizes to sites of UV-induced damage. We found that depletion of HBO1 resulted in UDS defects but maintained normal RRS. To understand the relationship between transcription or histone deacetylation-dependent and HBO1-dependent SNF2H recruitment at UV damage sites, we quantified HBO1 protein and mRNA expression after α -amanitin and TSA treatment. We discovered that both treatments strongly repressed HBO1 mRNA expression. As a consequence, HBO1 protein rapidly decreased to about 30% of control. Therefore, defects of SNF2H recruitment by α -amanitin and TSA reagents would partly cause of reduction of HBO1 protein. Regarding the contrasting results of UDS and RRS assays in HBO1- and SNF2H-depleted cells, we speculated that this is the result of HBO1-dependent and -independent SNF2H recruitments. The results of UDS assays in HBO1 depleted cells might indicate that acetylation of histones is more important for GG-NER than chromatin remodelling by ACF1-SNF2H. However, SNF2H would be supplied for a TC-NER HBO1-independent mechanism. ISWI is thought to bind near the transcription start site⁴⁴ and speculated to facilitate

an open chromatin structure for efficient CSB association²⁶. In general, histones of transcriptionally active genes are acetylated⁴⁵, and therefore SNF2H might be able to scan and bind to target nucleosome in the vicinity of lesions without HBO1.” (line 446-464)

16. Supplementary figure 6 should be included in the results section.

We have followed the reviewer’s suggestion. We moved previous Supplementary Fig. 6 to the results section as Fig. 7g.

17. It should be explained why different doses of UV are used in different IF experiments. Now, in Fig1a, 2d and 5a 12 J/m² is used (which seems rather low) and in fig. 1c-e is 50 J/m²

We have explained why we used different doses of UV in different IF experiments in the manuscript as follows:

“The sensitivity to detect proteins at damaged chromatin in cells fixed with MeOH/acetone was slightly lower compared with paraformaldehyde fixation-HCl treatment to detect CPD, although the MeOH/acetone method stably retained many kinds of proteins at damage sites, including XPC or HBO1 pS50/53. Even in experiments using different UV doses, we observed the same effects in our experiments (Supplementary Fig. 6c). Therefore, we used a high dose UV (50 J/m²) for some experiments.” (line 128-134)

Minor comments

1. line 54, "deficient GG-NER pathway results in predisposition to the autosomal recessive skin cancer, xeroderma pigmentosum". Please rephrase as xeroderma pigmentosum is not similar to recessive skin cancer. Skin cancer predisposition is one of the symptoms of XP.

We have changed this sentence as follows:

“A deficient GG-NER pathway results in xeroderma pigmentosum (XP). Predisposition to skin cancer is one of the symptoms of XP.” (line 67-69)

2. line 55-56, XPC is GG-NER initiator in eukaryotic cells, not only in mammalian cells. The authors state that XPC binding is to 'modify and relax the chromatin structure'. It is not clear on

what finding this statement is based on? Indeed XPC is the initiator of GG-NER, though in none of the provided references (1 and 2) no information is provided on chromatin modification/relaxation by XPC.

We have changed “mammalian” to “eukaryote” (line 69).

We agree with reviewer’s comment that references 1 and 2 provide no information on chromatin modification/relaxation by XPC. Therefore, we removed the phrase “to modify and relax the chromatin structure” from the sentence (line 70).

3. Given that the authors do not show that HBO1 itself is recruited (but phosphorylated at sites of UV damage), the title is misleading.

To address the reviewer’s comment, we have changed the title of this paper as follows:

“DDB2-dependent maintenance of phosphorylated HBO1 at UV irradiated sites is essential for nucleotide excision repair” (line 1-2)

4. line 60-68 As RBX1 is also part of the E3-ligase complex, it would be more appropriate and consistent (also with the authors previous paper) to refer to this complex as CRL4-DDB2 instead of CUL4/DDB2.

Thank you for this information. We added RBX1 in the text (line 75). We also changed the references of CUL4/DDB2 to CRL4/DDB2 in the manuscript.

5. lines 79-87 ALC1 is not the catalytic subunit of CHD even though its alternative name is CHD1L (=CHD1-like). CHD complexes have thus far not been found to be implicated in nucleotide excision repair. INO80 has been implicated (Jiang et al, PNAS 2010) and should be mentioned here as well. Furthermore, BRG1 was also implicated in mammalian NER by Zhang et al, Cell Cycle, 2009.

Thank you for these suggestions. We removed the phrase “the catalytic subunit of CHD” from the sentence (line 100). We have also mentioned the previous reports, Jiang et al, PNAS 2010

and Zhang et al, Cell Cycle, 2009, in the manuscript (line 102 and 99, respectively) (Ref. 25 and 23, respectively).

6. line 119. The authors state they have previously shown degradation of HBO1 starting from 2 h after UV. However, this is not detected in the experiment shown in Fig 1b. Do the authors still observe this degradation?

In our previous paper, we added cycloheximide (CHX) in the culture medium to inhibit new protein synthesis and we detected degradation of HBO1 starting from 2 h. However, in Fig. 1b of this paper, we did not add CHX in the culture medium, so the newly synthesized HBO1 was not phosphorylated by ATR. Only HBO1 pS50/53 was degraded in this experiment.

7. Line 188-189. The authors cannot draw the conclusion that XPC recruitment is facilitated by acetylation of H3 and H4 by HBO1, as they have not shown this directly. They only show a HBO1-dependent increase of H3/H4 acetylation levels after UV. Also, on page 11 the authors themselves already suggest that an additional function of HBO1 other than HAT activity may facilitate XPC recruitment, which contrast this conclusion drawn in lines 188-189.

We agree with the reviewer's comment. We have changed this sentence as follows:

“Based on these results, we concluded that UV-induced acetylation of H4 and H3 Lys14 around sites of UV lesions was HBO1-dependent.” (line 259-261)

8. Line 234, considering that the authors report involvement of SNF2H and ACF1 in NER, they should cite here that SNF2H and ACF1 were already found to be involved in transcription-coupled NER before (Aydin et al, NAR, 2014). Also, INO80 involvement should be cited (Jiang et al, PNAS 2010)

According to the reviewer's comment, we cited previous reports, Aydin et al, NAR, 2014 and Jiang et al, PNAS 2010, in the manuscript (Ref. 26 and 25, respectively).

9. Line 236 Please clarify what is meant with 'Our examination after local-UV irradiation'. Have the authors tested recruitment (to what type of damage?) of multiple ATP-dependent chromatin remodelers and found only ACF1 and SNF2H to be recruited?

We tested localization of BRM, BRG, ACF1 and SNF2H by immunostaining after local-UV irradiation (line 333-334).

10. Fig 1d Images are not equally sized

We moved previous Fig. 1d to Supplementary Fig. 1d and adjusted the sizes.

11. line 265 'physically interacts with SNF2H and recruits the ACF1-SNF2H complex to CPD sites'. The authors only show co-IP, although they do not provide evidence for direct physical interaction, can be through chromatin or other complex partners).

We agree with this comment. We removed the sentence “and recruited the ACF1-SNF2H complex to CPD sites” and “physically” from the manuscript. (line 368)

12. lines 296-299. HBO1-depleted cell could have a defect in overall GG-NER, i.e. also in the removal of 6-4PPs, which was not tested. The authors should mention this possibility.

We examined repair of 6-4PPs in HBO1-depleted cells in the revised manuscript (Supplementary Fig. 4a). We found that HBO1 was also required for repair of 6-4PPs. We have included this information in the new manuscript (line 270-274).

13. lines 302-307 How do the authors envision that ATM and/or ATR are activated after UV damage to phosphorylate HBO1? Is this happens during NER (and not because of DNA break induction in replicating cells), standard models predict that damage first needs to be processed by NER before ATR can be activated. How could ATM be activated? The authors should discuss this difficult to understand issue.

We have discussed our prediction on how ATR would be activated in G1 cells after UV irradiation in the Discussion section (line 483-490). We have included two new references in this discussion (Ref. 47 and 48).

Reviewer #2 (Remarks to the Author):

The manuscript by Hiroyuki Niida is a continuation of a previous work (it is cited within this work as well) where the authors have shown that HBO1 is predominantly phosphorylated by ATR and interacts with DDB2 upon exposure to UV irradiation. Here, the authors address the functional role of HBO1 in the repair of UV-induced DNA lesions. They first show that HBO1 is phosphorylated by ATR and that it requires DDB2 to recruit to UV-induced DNA lesions. They next show that HBO1 acetylates histone H3 Lys 14 and histone H4 aimed at facilitating the recruitment of XPC at sites of DNA damage (i.e. cyclobutane pyrimidine dimers) upon UV irradiation. In line, shHBO1 cells show delayed CPD repair and are sensitive to UV. The authors conclude that HBO1 plays a role during the early steps of the GG-NER subpathway in NER.

The data shown in this work are significant, of interest to the field, they are clearly presented and for the most part they seem to support the conclusions drawn in this work. The manuscript could benefit from further editing.

A major concern is that the authors do not provide adequate evidence for the specificity of HBO1 to UV-induced DNA CPDs as compared to i. other types of UV-induced DNA damages (6-4PPs), ii. to other types of DNA damages in general (DNA ICLs, DNA breaks, etc.) or even to iii. the TC-NER sub-pathway of NER. Supporting the specificity of their findings will further enhance the significance and scope of the work.

In response to the other types of UV-induced DNA damages (6-4PPs), we examined repair of 6-4PPs in HBO1-depleted cells. We discovered a defect in repair of 6-4PPs in HBO1-depleted cells. We have shown this data in **Supplementary Fig. 4a** and described it in the manuscript as follows:

“To examine this possibility, we measured CPD or 6-4PP repair efficiencies in HBO1-depleted cells (Fig. 5a and Supplementary Fig. 4a, respectively). As expected, depletion of HBO1 delayed the removal of CPDs and 6-4PPs from chromatin similar to the delay observed after depletion of DDB2 (Fig. 5a and Supplementary Fig. 4a, respectively).” (line 270-274)

1. Do the authors see a similar retention of HBO1 to chromatin when cells are exposed to the e.g. DNA cross-linker mitomycin (for DNA ICLs) or upon IR? It would be important to address the specificity of ph. HBO1 to UV-induced DNA damage.

Based on the reviewer's suggestion, we examined retention of HBO1 on the chromatin after mitomycin C (MMC) and IR treatment. We showed the results in Supplementary Fig. 4b and described details of this experiment in the manuscript as follows,

“We previously reported that Ser50 and 53 of HBO1 were phosphorylated in response to various DNA damage treatments¹⁵. Based on these observations, we examined whether HBO1 is involved in survival for ionizing radiation (IR) or mitomycin C (MMC) treatment, which induce DSB and DNA interstrand cross-link, respectively. We first monitored phosphorylation states of Ser50 and 53 in HBO1 (Supplementary Fig. 4b). Cells were irradiated with 10 Gy γ -irradiation or pre-treated with 50 ng/ml MMC for 1 h and then washed. Cells were cultured for indicated times and subjected to western blotting. Ser50 and 53 of HBO1 were immediately phosphorylated after γ -irradiation, and then phosphorylation was rapidly reduced (Supplementary Fig. 4b left panel). MMC treatment induced and maintained phosphorylation of Ser50 and 53 of HBO1 until 6 h after removal of MMC (Supplementary Fig. 4b right panel). Sensitivities of shHBO1 cells to IR and MMC were examined (Supplementary Fig. 4c). The sensitivity of shHBO1 cells to IR was almost same as with shCtl cells. Intriguingly, HBO1 depleted cells were sensitive for MMC treatment. These results suggested that HBO1 was required for not only UV but also MMC-induced DNA repair.” (line 284- 299)

2. The authors provide evidence that " Functional HBO1 is required for efficient GG-NER". The authors need to provide a clear justification and preferably experimental evidence on why they think the role of HBO1 in TC-NER is not relevant or that it should be excluded. Next to UDS and RNA synthesis recovery assay would be useful.

We appreciate the reviewer for this important question. We performed RNA synthesis recovery assays (RRS assays). We could not detect significant defects in HBO1-depleted HeLa and normal human fibroblasts or additional defects in UV^S-A fibroblasts depleted for HBO1. We showed these results in Fig. 2b and described this information in the text as follows:

“Next we performed recovery of RNA synthesis (RRS) assays to evaluate whether HBO1 was involved in TC-NER (Fig. 2b). Although depletion of CSB in HeLa cells showed defect in RRS, HBO1 depleted cells showed almost the same level of RRS compared with controls (Fig. 2b upper). *UVSSA* was identified as a one of the causative genes of UV-sensitive syndrome (*UV^SS*)²⁷. The cells derived from *UV^SS*-A patients are defective in TC-NER. Endogenous HBO1 or *UVSSA* was depleted by siRNAs (Fig. 2b). In contrast to reduction of RNA synthesis in *UVSSA*-depleted cells, cells depleted for HBO1 did not show any reduction of RNA synthesis after UV irradiation (Fig. 2b bottom). Furthermore, additional depletion of HBO1 in Kps3 fibroblasts derived from a *UV^SS*-A patient did not show any additive reduction of RNA synthesis.” (line 183-192)

3. The blots on Ac-H4 are not clear. If possible some clearer, higher quality images should be provided.

We re-probed aliquots of the remaining samples with a new lot of anti-acetyl-Histone H4 antibody and have replaced the previous blots with new ones. (Fig. 4a)

4. Are shHBO1 cells sensitive to other type of DNA damages (besides UV)?

To answer the reviewer’s question, we performed clonogenic assay after MMC and IR treatments. We found that shHBO1 cells had a similar sensitivity to IR. Interestingly, shHBO1 cells were more sensitive to MMC treatment. We showed these results in **Supplementary Fig. 4c** and described the results in the manuscript as follows:

“Based on these observations, we examined whether HBO1 is involved in survival for ionizing radiation (IR) or mitomycin C (MMC) treatment, which induce DSB and DNA interstrand cross-link, respectively. We first monitored phosphorylation states of Ser50 and 53 in HBO1 (Supplementary Fig. 4b). Cells were irradiated with 10 Gy γ -irradiation or pre-treated with 50 ng/ml MMC for 1 h and then washed. Cells were cultured for indicated times and subjected to western blotting. Ser50 and 53 of HBO1 were immediately phosphorylated after γ -irradiation, and then phosphorylation was rapidly reduced (Supplementary Fig. 4b left panel). MMC treatment induced and maintained phosphorylation of Ser50 and 53 of HBO1 until 6 h after removal of MMC (Supplementary Fig. 4b right panel). Sensitivities of shHBO1 cells to IR and MMC were examined (Supplementary Fig. 4c). The sensitivity of shHBO1 cells to IR was

almost same as with shCtl cells. Intriguingly, HBO1 depleted cells were sensitive for MMC treatment. These results suggested that HBO1 was required for not only UV but also MMC-induced DNA repair.” (line 285-299)

5. Could the authors use separate inhibitors to distinguish whether HBO1 role requires ATM or ATR specifically? Otherwise they should provide a justification why when ATR was shown to predominantly phosphorylate HBO1, the authors use an ATR/ATM inhibitor.

Based on the reviewer’s suggestion, we investigated the effects of specific inhibitors against ATM or ATR on phosphorylation of Ser50 and 53 of HBO1. We found the ATR inhibitor alone significantly inhibited phosphorylation of HBO1 at the sites of UV lesions. We showed these results in Fig. 1c.

6. Do CSB or CSA cells show (or not) additional UV sensitivity following depletion of HBO1?

To answer the reviewer’s question, we established stably depleted UV^SS-A fibroblasts and compared the UV sensitivities between UV^SS-A fibroblasts and UV^SS-A with shHBO1 fibroblasts. We detected additional UV sensitivity in UV^SS-A with shHBO1 fibroblasts. We showed this result in Fig. 6c. We further tested this point using HeLa cells depleted for CSB, HBO1 and CSB/HBO1. Consistent with the results in UV^SS-A with shHBO1 fibroblasts, we detected additional UV sensitivity in double-depleted cells. We have shown this result in Supplementary Fig. 5b. We described these results in the manuscript as follows:

“Enhanced sensitivity toward UV irradiation after depletion of HBO1 was evaluated in normal, XP and UV^SS-A fibroblasts (Fig. 6c). Although normal fibroblasts became more sensitive after depletion of HBO1, XP-E, XP-C and XP-A cells did not show additional UV sensitivity following depletion of HBO1. Consistent with UDS and RRS assays (Fig. 2), depletion of HBO1 from UV^SS-A fibroblasts induced additional UV sensitivity (Fig. 6c). The same additional sensitivity was observed in CSB and HBO1 double-depleted cells (Supplementary Fig. 5b).” (line 317-323)

7. The authors use γ H2AX as an indicator of DSB sites. As such they assume and subsequently show no γ H2AX foci in UV-treated cells. This is somewhat surprising as γ H2AX foci are known to be detected even in 10% of non-irradiated cells and do increase remarkably upon UV

irradiation (several papers exist). The authors need to re-consider where they stand by their original observation.

We apologize for the low sensitivity of our previous immunostaining with anti- γ H2AX. To show that UV irradiation did not generate DSB, we quantified fluorescence of γ H2AX after ATR or ATM/DNA-PK treatment. ATM and DNA-PK are well known kinases that are activated by DSB. When ATM/DNA-PK inhibitors were applied to the medium, reduction of fluorescence of γ H2AX was not observed. Only the ATR inhibitor suppressed fluorescence of γ H2AX (Supplementary Fig. 6b). Therefore, we concluded that 20 J/m² UV irradiation did not generate DSB and recruitment of ACF1 and SNF2H at the sites of UV lesions were not responsive to DSB.

Reviewers' comments:

Reviewer #2 (Remarks to the Author):

Comments on the author's rebuttal:

1. It is also not correct to write that HBO1 is 'maintained' at the site of damage. This phrasing suggests that there is an active mechanism keeping HBO1 at damaged sites, for which the authors provide no proof. The usage of 'maintain' now leads to strange, illogical sentences such as 'we speculated that HBO1 would remain at UV-damaged sites in a DDB2-dependent manner' (line 116). The only observation is that HBO1 is phosphorylated at the site of damage. This is what should be described. Only the additional sentences in the discussion (lines 477-481) correctly describe this. The authors now also include the chromatin fraction in Fig 1b, as requested.
2. Image quality is better now. Also, as requested, control damage staining is provided and signals quantified. It is, however, confusing that the authors switch from quantifying co-localization (for instance fig. 1c and 3e and g) to quantifying relative fluorescence (for instance fig 1d). Co-localization is based on fluorescence intensity, so it would be more logical and less confusing to always show the fluorescent intensity. Now, it is unclear how co-localization is defined by the authors. Does the absence of co-localization mean that no signal was observed by eye or that no signal (higher than overall nuclear background) was observed by quantification? Also, it is unclear how relative fluorescence is defined. For instance in Fig 1d, is the 'relative fluorescence of XPC/DDB2' the XPC signal divided by the DDB2 signal, is it XPC signal divided by XPC nuclear signal or is it absolute intensity of XPC signal expressed in percentage? The same holds for other figures showing relative fluorescence. The authors should clarify their quantifications.
3. The authors have appropriately addressed our concern.
4. The authors have appropriately addressed our concern.
5. The authors now show that 405 nm laser with Hoechst 33342 induces CPDs that activate the NER pathway. Without the addition of Hoechst33342 no CPD formation is observed, rendering the previous result with GFP-SNF2H invalid. Strikingly, after performing the experiment with GFP-SNF2H the right way, the authors get the same results. The authors should provide a satisfactory explanation for this peculiar observation and striking contradiction. In the materials and methods it is still stated that the experiment was performed without Hoechst 33342. This should be corrected.
6. See comment 2. It would be less confusing if the authors just show relative XPC levels in Fig 3e and g, for the 20 min and 60 min time points. These results, because of which the authors argue that XPC is retained longer at damage sites after HBO1 knockdown, are at present not convincing. The significance of the differences shown in Fig 3E and G is unclear, as comparing these figures shows that shCtl XPC co-localization in Fig 3G is at the same level as shDDB2 and shHBO1 in Fig 3E. These differences could just as well be attributed to experimental variation. It would be more convincing and less confusing if the authors could show that the relative fluorescence levels of XPC are lower at 20 min and higher at 60 min after HBO1 knockdown.
7. On the blot shown in Fig 4a, Ach4 peaks immediately at 20 min which is different from the previous experiment shown where Ach4 peaks much later. Is this a new experiment? The authors now show DDB2, but was this also taken into account in the quantification (as it should, since this represents the amount of IP-ed chromatin by this procedure; IP by anti-DDB2), because the quantification is exactly similar as in the previous version of the manuscript.
The timing of the experiment shown in Fig 4b is rather strange as the experiment is performed 2 h after UV damage (how much UV was used?), after this time XPC recruitment to local damaged sites is already over its peak (as a significant fraction of 6-4PP lesions are removed by that time, which were shown to be the lesions to which the vast majority of XPC is recruited, see Fig 3e). So the authors cannot conclude, as they do in lines 265-266 that this acetylation facilitates recruitment of XPC. Also, it is not convincing to compare this experiment with that in Fig 4c to claim that HBO1 acetylates H3K14 at sites of damage. To properly be able to compare conditions before and after UV, the authors should perform both IPs in the same experiment, preferably blotted on the same western and at an earlier time point after UV. Moreover, the lower reduction of H3AcK14 in Fig 4c as compared to Fig 4b, claimed to be caused by the absence of UV -

irradiation, could equally be caused by lower reduction of HBO1.

8. The authors have appropriately addressed our concern.

9. Remarkably, the authors only removed the highest UV dose (60 J/m²) point from the graphs. It still remains that the observed fraction of surviving cells at the indicated doses does not match with UV-survival data in the literature (too much examples to list here).

10. The authors have now explained in the figure legends that the '0 h' after UV actually means 'no UV'. This should be indicated in the figures as well. Additionally, it is unclear why CPD levels immediately after UV were not determined, as this would show whether similar amounts of damage are initially induced despite HBO1 depletion. Now, the first time point to determine damage levels is after 1 h (for CPDs; for 64PP in Fig S4 this is 30 min), but given the fact that the authors observe effects of HBO1 depletion on XPC immediately after damage induction, the repair immediately after UV induction should also be taken into account. Therefore, initial damage levels immediately after UV should be determined.

11. The authors have appropriately addressed our concern. In addition, they show that α -amanitin and TSA treatment reduce HBO1 levels, which could partly explain diminished SNF2H UV-damage accumulation in Aydin et al. 2014.

12. The point of our concern was not that the HA-DDB2 panel should be exposed longer. The point was that Myc-HBO1 is hugely overexpressed and that only a very minor fraction (less than 5-10%) of Myc-HBO1 is IPed with GFP-SNF2H. Therefore, this reciprocal IP does not suggest that SNF2H and HBO1 significantly interact.

13. The authors have appropriately addressed our concern.

14. The authors have appropriately addressed our concern. For most new si/shRNAs similar effects are observed. However, with new siSNF2H (siSNF2H-2) in Supplementary Fig. 8b hardly any effect on XPC-eGFP is observed in spite of very efficient SNF2H knockdown. The authors should mention or discuss this.

15. The authors have appropriately addressed our concern.

16. The authors have appropriately addressed our concern.

17. The authors have addressed our concern. However, even if the same effects are observed with different doses of UV, it is still unclear why the authors use 12 J/m² in one experiment and 50 J/m² in other experiments. What is the rationale for using different UV doses? It is only confusing. Minor comments

All minor comments were appropriately addressed, except for comment 3 and 13. The title is still incorrect, because the authors do not show maintenance, but induction of HBO1 phosphorylation. The title should be adjusted. As for comment 13, the authors now provide some discussion on how ATR might be activated. However, their idea that ATR activation involves an interaction of TopBP1 with N-Aco-AAF is strange because N-Aco-AAF is not used in any experiment. TopBP1 can indeed activate ATR but this involves replication stress and thus replicating cells. The second possibility (mentioned starting in line 489) seems more feasible, but the authors should explain (with proper references) how 'the first NER' activates ATR, i.e. involving processing of the lesion by NER and damage signaling activating ATR due to the resulting NER-dependent single strand break or single-strand gap induced by the dual incision. The provided explanation would benefit from English proof reading. This model would, however, predict that in the absence of functional NER, ATR will not be activated (by 'the first NER') and HBO1 will not be phosphorylated. This contrasts the observation of the authors that in NER deficient cells, such as XP3KA and XP7HM (Fig 6A), HBO1 is still phosphorylated. Can the authors come up with better explanations, that accord with standard models in literature, how ATR could be activated?

Additional comments

- Figure 4b and c, the authors quantify AchH4 and H3AcK14 levels by dividing signals by H4 and H3 levels. However, H4 (first panel fig 4b) and H3 (first panel 4c) signals are vague and not suited for quantification. How did the authors perform these quantifications?

- line 72, 'chromatin' should be 'DNA'

- line 95, ATP-dependent chromatin remodeling is not required for changing chromatin structure by definition. Chromatin structure can also change without ATP-dependent chromatin remodeling. This should be phrased correctly.

- line 100, explain that ALC1 is an ATP-dependent chromatin remodeler, but it is not part of any of the four chromatin remodeling complexes mentioned in line 96
- line 123, 'hardly detected' should be 'not detected' (unless the authors provide evidence that there is some accumulation).
- overall, the authors should clearly indicate in each legend and figure the amount of UV damage induced and the time after UV at which the experiment was performed. This is not clear for all figures. In addition, it is not clear for all figures which cell lines were used. Are these in all cases HeLa, unless otherwise indicated? This should be made more clear.
- Lines 134-137. It is unclear why the authors speculate that HBO1 pS50/53 binding to damaged chromatin was decreased because of hydrochloric treatment, as they state that the sensitivity in MeOH/acetone treated samples is lower. Please clarify in the text.
- line 231, 70% is incorrect judging from fig 3e.
 - line 236, one can hardly call co-localization of 30-40% 'abundant'. The co-localization level is slightly above shCtl (and similar to the shCtl of fig 3g!!).
 - line 287, IR induces oxidative damage, single strand and some double strand breaks. MMC induces both inter- and intrastrand crosslinks. Intra-strand crosslinks are repaired by NER. This should be mentioned and can also explain the HBO1-dependent sensitivity the authors observe for MMC.
 - line 346, Figs 7A and S6C, if the authors wish to show that ACF1 (and SNF2H) do not respond to DSBs induced by UV irradiation, they should perform these local UV damage experiments in non-replicating cells. In replicating cells, UV damage can induce DSBs and/or replication stress that activates ATR, which subsequently phosphorylates H2AX. To observe that H2AX phosphorylation after UV is dependent on ATR, therefore does not prove that in the local damage shown in Figs 7A and S6C no double strand breaks or replication stress is induced. Also, the authors should indicate at which time point after UV these local damage experiments were performed.
 - line 428, HBO1 phosphorylation appears already saturated 30 min after UV in Fig 1b
 - line 434, maximum loading of XPC appears is already evident at 30 min after UV in Fig 1b.
 - line 445-446, UV should not be listed as genotoxic agent that induces (only) DSBs. It would be better to speak of genotoxic agents that induce (single and double) strand breaks, oxidative damage, replication stress and helix-distorting DNA damage.
- English style and grammar could be improved throughout the manuscript

Reviewer #3 (Remarks to the Author):

The authors have answered my previous remarks and I have no further comments.

Response to Reviewer #2 (Remarks to the Author):

We thank the reviewer for many constructive comments, which we have helped us to greatly improve our paper. Our specific responses to the comments are included below.

Comments on the author's rebuttal:

1. It is also not correct to write that HBO1 is 'maintained' at the site of damage. This phrasing suggests that there is an active mechanism keeping HBO1 at damaged sites, for which the authors provide no proof. The usage of 'maintain' now leads to strange, illogical sentences such as 'we speculated that HBO1 would remain at UV-damaged sites in a DDB2-dependent manner' (line 116). The only observation is that HBO1 is phosphorylated at the site of damage. This is what should be described. Only the additional sentences in the discussion (lines 477-481) correctly describe this. The authors now also include the chromatin fraction in Fig 1b, as requested.

We apologize for using the incorrect wording ("maintained") in describing HBO1 at the site of damage. As the reviewer pointed out, our data suggested that HBO1 present on chromatin is phosphorylated in response to UV-induced DNA damage. Therefore, we have revised the wording throughout the manuscript to better reflect our intended statements as follows.

First, we have revised the title from "DDB2-dependent maintenance of phosphorylated HBO1 at UV irradiated sites is essential for nucleotide excision repair" to "Phosphorylated HBO1 at UV irradiated sites is essential for nucleotide excision repair".

We also revised this sentence as shown below:

"We examined whether phosphorylated HBO1 is present at UV damaged sites." (line 112-113)

We also revised other sections in the manuscript, including the following:

"Here, we show that phosphorylated HBO1 at cyclobutane pyrimidine dimer (CPD) sites mediates histone acetylation to facilitate recruitment of XPC at the damaged DNA sites." (line 41-43)

“HBO1 is phosphorylated at CPD sites by ATR.” (line 109)

“Surprisingly, accumulations of both ACF1 and SNF2H at DNA damage sites were decreased in shDDB2 and shHBO1 cells (Fig. 7a, b and Supplementary Fig. 6b).” (line 340-342)

“Here we demonstrate for the first time that HBO1 is involved in the early step of CPD repair in GG-NER.” (line 409-410)

“HBO1 was phosphorylated at sites of local UV irradiation.” (line 875)

“Fig. 7 The ACF1-SNF2H chromatin remodeller interacts with HBO1 and accumulates at UV-damaged DNA.” (line 990-991)

2. Image quality is better now. Also, as requested, control damage staining is provided and signals quantified. It is, however, confusing that the authors switch from quantifying co-localization (for instance fig. 1c and 3e and g) to quantifying relative fluorescence (for instance fig 1d). Co-localization is based on fluorescence intensity, so it would be more logical and less confusing to always show the fluorescent intensity. Now, it is unclear how co-localization is defined by the authors. Does the absence of co-localization mean that no signal was observed by eye or that no signal (higher than overall nuclear background) was observed by quantification? Also, it is unclear how relative fluorescence is defined. For instance in Fig 1d, is the ‘relative fluorescence of XPC/DDB2’ the XPC signal divided by the DDB2 signal, is it XPC signal divided by XPC nuclear signal or is it absolute intensity of XPC signal expressed in percentage? The same holds for other figures showing relative fluorescence. The authors should clarify their quantifications.

We appreciate the reviewer’s helpful comments and suggestions. We agree that the use of fluorescence intensity is more logical. In our previous manuscript, we judged co-localization of signals by eye. Our definition of relative intensity of local UV

irradiation is described in Supplementary Fig. 1b. We have used the terminology “relative fluorescence” in describing fluorescence at local UV irradiated sites. We used the wording “relative XPC-EGFP accumulation” and “relative GFP-SNF2H accumulation” for the live cell imaging experiments (Fig. 3b, c, d, Fig. 7c, f, Supplementary Fig. 3e, Supplementary Fig. 8b and c). We used the wording “pS50/53/cell” or “ γ H2AX/cell” in experiments that quantified single fluorescence immunostaining (Fig. 1d, Supplementary Fig. 1d and Supplementary Fig. 5b). In the revised manuscript, we have removed all co-localization figures and now present figures that show relative fluorescence.

Importantly, we found that the relative fluorescence of pS50/53 HBO1 (normalized to CPD fluorescence) in shXPC cells (Fig. 1a) and in XPC and XPA fibroblasts (Fig. 6a) was lower than that in control cells. We speculated that phosphorylations of HBO1 at G1 phase might be deficient in shXPC HeLa cells and XPC and XPA fibroblasts, because a previous report stated that the NER pathway was necessary for γ H2AX by ATR (O'Driscoll, M *et al.* 2003). Therefore, we compared the phosphorylation of pS50/53 HBO1 and γ H2AX at G1 phase in shCtl and shXPC cells using the IN CELL analyzer. As we expected, phosphorylations of pS50/53 HBO1 and γ H2AX at G1 were decreased in shXPC cells. We have presented these new results in Fig. 1d. We also compared the phosphorylations of pS50/53 HBO1 in normal human dermal fibroblasts (NHDF) and XPA fibroblasts (XP7HM) and show these results in Supplementary Fig. 5b. We apologize as we have retracted the previous supplementary Fig. 1c that showed same relative intensities of pS50/53 HBO1 between shCtl and shXPC-2. We suspect that most of the analyzed cells in our previous experiments were except G1 phase.

Again, we thank the reviewer for the important comments, which have helped us to improve our paper.

3. The authors have appropriately addressed our concern.

4. The authors have appropriately addressed our concern.

5. The authors now show that 405 nm laser with Hoechst 33342 induces CPDs that active the NER pathway. Without the addition of Hoechst33342 no CPD formation is

observation, rendering the previous result with GFP-SNF2H invalid. Strikingly, after performing the experiment with GFP-SNF2H the right way, the authors get the same results. The authors should provide a satisfactory explanation for this peculiar observation and striking contradiction. In the materials and methods it is still stated that the experiment was performed without Hoechst 33342. This should be corrected.

We apologize for providing contradictive results regarding GFP-SNF2H in the first version of the manuscript. After the first review by this reviewer, we examined generation of CPD and accumulation of GFP-XPA at the 405-nm laser irradiated areas with and without Hoechst33342. As shown in the attached file “For reviewer #2a”, we could detect accumulation of GFP-XPA at the 405-nm laser irradiated areas with Hoechst33342 but could not detect accumulation of GFP-XPA without Hoechst33342. Therefore, we evaluated the accumulation of GFP-SNF2H in shDDB2 and shHBO1 cells in the presence of Hoechst33342. Although some reports have shown generation of DSB by 405-nm laser irradiation in the presence of Hoechst33342 and accumulation of ACF1/SNF2H at DSB sites, we observed a reduction of GFP-SNF2H accumulation in shDDB2 and shHBO1 cells. During the latest revision, we obtained similar results regarding reduction of GFP-SNF2H accumulation in the presence and absence of Hoechst33342 in shHBO1 cells. To determine the reproducibility of these two results, we repeated experiments that monitor the accumulation of GFP-SNF2H in the presence and absence of Hoechst33342 at the same time. As shown in the attached file “For reviewer#2b,c”, we could detect reduced accumulation of GFP-SNF2H in shHBO1 cells in the presence of Hoechst33342. The difference of GFP-SNF2H accumulation between shCtl and shHBO1 was statistically significant. However we could not detect any significant accumulation of GFP-SNF2H in the absence of Hoechst33342. Therefore, we deeply apologize as we are retracting the first GFP-SNF2H data because the experiment was not reproducible. We would like to present the GFP-SNF2H accumulation in the presence of Hoechst33342 in our paper (Fig 7c). Although not only CPD but also DSB are generated by this condition, decreased accumulation of GFP-SNF2H in shDDB2 and shHBO1 cells was still detected. We speculate that accumulation of GFP-SNF2H in response to CPD was suppressed by depletion of DDB2 or HBO1. We have described our explanation in the manuscript as follows:

“We further examined SNF2H accumulation by monitoring GFP-SNF2H in living cells (Fig. 7c). Although 405-nm laser irradiation with Hoechst 33342 in the culture medium induces not only CPDs but also DSBs in DNA³⁸, we found reduced accumulation of GFP-SNF2H in shDDB2 and shHBO1 cells (Fig. 7c, Supplementary Fig. 6c). These observations suggest that DDB2 and HBO1 facilitate accumulation of the SNF2H complex at CPD sites.” (line 342-347)

We also presented typical imaging of the reduced accumulation of GFP-SNF2H in shDDB2 and shHBO1 cells in Supplementary Fig. 6c.

We corrected the methodology for our GFP-SNF2H analysis, which was not corrected during the previous revision. (line 610)

6. See comment 2. It would be less confusing if the authors just show relative XPC levels in Fig 3e and g, for the 20 min and 60 min time points. These results, because of which the authors argue that XPC is retained longer at damage sites after HBO1 knockdown, are at present not convincing. The significance of the differences shown in Fig 3E and G is unclear, as comparing these figures shows that shCtl XPC co-localization in Fig 3G is at the same level as shDDB2 and shHBO1 in Fig 3E. These differences could just as well be attributed to experimental variation. It would be more convincing and less confusing if the authors could show that the relative fluorescence levels of XPC are lower at 20 min and higher at 60 min after HBO1 knockdown.

We agree with this reviewer’s suggestion and removed the previous co-localization figures (previous Fig. 3e and g). We measured the relative fluorescence of XPC at local UV irradiated areas at 20 and 60 min. Relative fluorescence was shown as a percentage of shCtl cells at 20 min. As the reviewer predicted, in shHBO1 and shDDB2 cells, the relative fluorescence of XPC at 60 min was higher than that at 20 min. In contrast, in shCtl cells, the relative fluorescence of XPC at 20 min was higher than that at 60 min (Fig. 3f). ShHBO1 cells stably expressing SA and EQ mutants showed the same delayed accumulation of XPC (Fig. 3g).

7. On the blot shown in Fig 4a, Ach4 peaks immediately at 20 min which is different

from the previous experiment shown where AcH4 peaks much later. Is this a new experiment? The authors now show DDB2, but was this also taken into account in the quantification (as it should, since this represents the amount of IP-ed chromatin by this procedure; IP by anti-DDB2), because the quantifications is exactly similar as in the previous version of the manuscript.

We had changed the AcH4 panel to follow another reviewer's request in the second version of our paper. However, as the current reviewer suggested, we had not quantified relative acetylation of AcH4 including new AcH4 western blotting. We apologize and have re-quantified the relative AcH4 levels including new panel. We found that the relative acetylation of AcH4 at 20 min was significantly increased by two- to three-fold of AcH4 levels in non-irradiated cells (Fig. 4a).

The timing of the experiment shown in Fig 4b is rather strange as the experiment is performed 2 h after UV damage (how much UV was used?), after this time XPC recruitment to local damaged sites is already over its peak (as a significant fraction of 6-4PP lesions are removed by that time, which were shown to be the lesions to which the vast majority of XPC is recruited, see Fig 3e). So the authors cannot conclude, as they do in lines 265-266 that this acetylation facilitates recruitment of XPC. Also, it is not convincing to compare this experiment with that in Fig 4c to claim that HBO1 acetylates H3K14 at sites of damage. To properly be able to compare conditions before and after UV, the authors should perform both IPs in the same experiment, preferably blotted on the same western and at an earlier time point after UV. Moreover, the lower reduction of H3AcK14 in Fig 4c as compared to Fig 4b, claimed to be caused by the absence of UV-irradiation, could equally be caused by lower reduction of HBO1.

We performed ChIP-Western blotting analysis using cells irradiated with 40 J/m² UV for 0 min, 20 min and 120 min and indicated these conditions (time points and UV dose) in the manuscript (line 235) and the figure legend for new Fig. 4a. As shown in Fig. 4a, we detected a time-dependent increase of histone H3K14Ac and AcH4. Because we did not measure retention of XPC at 120 min after UV 40 J/m², we retracted our conclusion pointed out by the reviewer (previous line 265-266).

Regarding the previous Fig. 4c, we agree with the reviewer's comment. We performed

both IPs in the same experiment and blotted on the same western blot at 20 min after UV irradiation (Fig. 4b). The results indicated that both AcH4 and H3K14Ac are enhanced only in shCtl cells after UV irradiation, whereas depletion of HBO1 inhibited the UV-induced acetylation events. These results suggest that acetylation of histone H4 and H3K14 around UV lesions is catalyzed by HBO1.

8. The authors have appropriately addressed our concern.

9. Remarkably, the authors only removed the highest UV dose (60 J/m²) point from the graphs. It still remains that the observed fraction of surviving cells at the indicated doses does not match with UV-survival data in the literature (too much examples to list here).

According to the reviewer's comment, we performed these clonogenic assays with low-dose UV irradiation in primary fibroblasts with or without endogenous HBO1 depletion (Fig. 6c). The sensitivities of normal and XP fibroblasts to low-dose UV irradiation are similar to those published in previous report (Niedernhofer LJ *et al.* 2006). We found that depletion of HBO1 increased the UV sensitivity in normal and UVSSA fibroblasts. We have replaced the figure with these results.

10. The authors have now explained in the figure legends that the '0 h' after UV actually means 'no UV'. This should be indicated in the figures as well. Additionally, it is unclear why CPD levels immediately after UV were not determined, as this would show whether similar amounts of damage are initially induced despite HBO1 depletion. Now, the first time point to determine damage levels is after 1 h (for CPDs; for 64PP in Fig S4 this is 30 min), but given the fact that the authors observe effects of HBO1 depletion on XPC immediately after damage induction, the repair immediately after UV induction should also be taken into account. Therefore, initial damage levels immediately after UV should be determined.

We have clarified that the time point '0 h' actually means 'no UV' in the figures (as indicated by "-UV"). To follow the reviewer's suggestions, we performed CPD repair assays at 0 h, 1 h, 8 h, and 24 h after damage induction and without UV (Fig. 5a, b and Fig. 6b). We also reexamined the 6-4PP repair assays at 0 h, 0.5 h, and 4 h after damage

induction and without UV as shown in Supplementary Fig. 4a. Both CPD and 6-4PP removals at early time points (1 h and 0.5 h, respectively) were observed in both HeLa cells and normal human fibroblasts without HBO1 depletion (Fig. 5a, b, Fig. 6b and Supplementary Fig. 4a).

11. The authors have appropriately addressed our concern. In addition, they show that α -amanitin and TSA treatment reduce HBO1 levels, which could partly explain diminished SNF2H UV-damage accumulation in Aydin et al. 2014.

12. The point of our concern was not that the HA-DDB2 panel should be exposed longer. The point was that Myc-HBO1 is hugely overexpressed and that only a very minor fraction (less than 5-10%) of Myc-HBO1 is IPed with GFP-SNF2H. Therefore, this reciprocal IP does not suggest that SNF2H and HBO1 significantly interact.

We agree with the reviewer's suggestion that only a minor fraction of Myc-HBO1 was co-immunoprecipitated with GFP-SNF2H. We reexamined the experiment using anti-GFP-conjugated agarose beads (MBL) to improve the immunoprecipitation of GFP-SNF2H. As shown in Fig. 7d, co-immunoprecipitated Myc-HBO1 but not HA-DDB2 was detected in the anti-GFP bead immunoprecipitates. However, the co-immunoprecipitation of Myc-HBO1 with GFP-SNF2H was still low, which indicated that the interaction between GFP-SNF2H and Myc-HBO1 is not strong. We then also evaluated the interaction between endogenous HBO1 and endogenous SNF2H. As shown in Fig. 7e, we found that endogenous HBO1 interacted with endogenous SNF2H in HeLa cells. These results confirm that HBO1 can interact with SNF2H in intact cells.

We have included the IP methodology in the Materials and Methods section (line 514-515).

We revised the Results section as follows:

“we investigated whether HBO1 interacted with SNF2H. As shown in Fig. 7d, GFP-SNF2H successfully co-immunoprecipitated Myc-HBO1 but not HA-DDB2. However, we observed that despite high expression levels of Myc-HBO1, the amount of Myc-HBO1 that co-immunoprecipitated with GFP-SNF2H was low, suggesting that the

interaction between GFP-SNF2H and Myc-HBO1 is not strong. Thus, we next evaluated the interaction between endogenous HBO1 and endogenous SNF2H. As shown in Fig. 7e, we found that endogenous HBO1 interacted with endogenous SNF2H in HeLa cells. These results suggest that HBO1 can interact with SNF2H in intact cells.” (line 349-357)

13. The authors have appropriately addressed our concern.

14. The authors have appropriately addressed our concern. For most new si/shRNAs similar effects are observed. However, with new siSNF2H (siSNF2H-2) in Supplementary Fig. 8b hardly any effect on XPC-eGFP is observed in spite of very efficient SNF2H knockdown. The authors should mention or discuss this.

To follow the reviewer’s suggestion, we mentioned the weak effect on XPC-EGFP in cells treated with siSNF2H-2 in the Results. We also examined whether depletion of SNF2H by siSNF2H-2 was able to decrease accumulation of XPC-EGFP and found a significant reduction of XPC-EGFP accumulation in siSNF2H-2-treated cells. We showed this result as **Supplementary Fig. 8c** and included the following text:

“Live cell imaging experiment showed that accumulation of XPC-EGFP was partially compromised in SNF2H-depleted cells compared with HBO1-depleted cells; however, we observed minor defects in XPC-EGFP accumulation in cells depleted for SNF2H by SNF2H-2 siRNA (Supplementary Fig. 8b, graph, green line). To clarify whether depletion of SNF2H reduces accumulation XPC-EGFP, we repeated SNF2H depletion by siSNF2H-2 and examined its effect on XPC-EGFP accumulation (Supplementary Fig. 8c), and found that accumulation of XPC-EGFP was significantly decreased in SNF2H-depleted cells. In addition, double depletion of SNF2H and HBO1 did not show any additional defect in XPC-EGFP accumulation compared with HBO1 single depletion (Fig. 7f).” (line 379-389)

15. The authors have appropriately addressed our concern.

16. The authors have appropriately addressed our concern.

17. The authors have addressed our concern. However, even if the same effects are observed with different doses of UV, it is still unclear why the authors use 12 J/m² in one experiment and 50 J/m² in other experiments. What is the rationale for using different UV doses? It is only confusing.

We apologize for using different doses of UV. We will use consistent irradiation doses in our future studies.

Minor comments

All minor comments were appropriately addressed, except for comment 3 and 13. The title is still incorrect, because the authors do not show maintenance, but induction of HBO1 phosphorylation. The title should be adjusted.

As mentioned our response to comment 1, we have revised the title to “Phosphorylated HBO1 at UV irradiated sites is essential for nucleotide excision repair”.

As for comment 13, the authors now provide some discussion on how ATR might be activated. However, their idea that ATR activation involves an interaction of TopBP1 with N-Aco-AAF is strange because N-Aco-AAF is not used in any experiment. TopBP1 can indeed activate ATR but this involves replication stress and thus replicating cells. The second possibility (mentioned starting in line 489) seems more feasible, but the authors should explain (with proper references) how ‘the first NER’ activates ATR, i.e. involving processing of the lesion by NER and damage signaling activating ATR due to the resulting NER-dependent single strand break or single-strand gap induced by the dual incision. The provided explanation would benefit from English proof reading. This model would, however, predict that in the absence of functional NER, ATR will not be activated (by ‘the first NER’) and HBO1 will not be phosphorylated. This contrasts the observation of the authors that in NER deficient cells, such as XP3KA and XP7HM (Fig 6A), HBO1 is still phosphorylated. Can the authors come up with better explanations, that accord with standard models in literature, how ATR could be activated?

We understand that TopBP1 involves replication stress to activate ATR. Therefore, we removed the sentences describing activation of ATR via TopBP1.

In response to reviewer's comment 2, we measured the relative intensities of pS50/53 (Fig. 1a and Fig. 6a). Quite interestingly, the relative intensities of pS50/53 HBO1 were lower in shXPC HeLa cells (Fig. 1a) and XP3KA and XP7HM fibroblasts compared with control cells (Fig. 6a). These results suggested that the NER pathway is required for phosphorylations of HBO1.

To clarify whether the first NER is required for phosphorylation of HBO1 during G1 phase, we quantified pS50/53 at G1 phase in shCtl and shXPC cells (Fig. 1d). The results indicated that the phosphorylations of HBO1 were significantly decreased only in G1 but not in S-G2/M phase in shXPC cells. Consistent with this finding and the previous report by O'Driscoll *et al.* (ref no. 27), γ H2AX was also significantly reduced at G1 in shXPC cells. We also quantified phosphorylations of HBO1 in XP7HM fibroblasts at G1 phase by the IN Cell analyzer and found that pS50/53 were significantly decreased at G1 phase in XP7HM cells (Supplementary Fig. 5b).

We also performed local UV irradiation experiments in normal and XP7HM fibroblasts synchronized at G0 (G0) or asynchronous (AS) (Supplementary Fig. 5a). In contrast to the detectable levels of pS50/53 at UV irradiated areas in G0 normal fibroblasts, pS50/53 was hardly detected in G0 XP7HM fibroblasts. Importantly, pS50/53 were detected at similar intensities in some parts of AS XP7HM fibroblasts. Representative images of local UV irradiation experiments of G0 and AS XP7HM cells are shown in Supplementary Fig. 5a. Based on these results, we predict that pS50/53 positive XP3KA (XPC) and XP7HM (XPA) cells were in S phase and could activate ATR without the NER pathway.

Additional comments

- Figure 4b and c, the authors quantify AcH4 and H3AcK14 levels by dividing signals by H4 and H3 levels. However, H4 (first panel fig 4b) and H3 (first panel 4c) signals are vague and not suited for quantification. How did the authors perform these quantifications?

We agree with this comment. We had previously used ImageJ to quantitate the signals, although we agree that these are vague signals. Therefore, we repeated these

experiments with more cells and used a more sensitive ECL reagent (RPN2235, GE Healthcare). These results showed stronger signals and are presented as Fig. 4b. Signal intensities were quantitated using ImageJ. Ratios of AcH4/H4 or H3AcK14/H3 from five independent experiments were statistically evaluated by Student's t test (Fig. 4 legends).

- line 72, 'chromatin' should be 'DNA'

Thank you very much. We changed "chromatin" to "DNA". (line 69)

- line 95, ATP-dependent chromatin remodeling is not required for changing chromatin structure by definition. Chromatin structure can also change without ATP-dependent chromatin remodeling. This should be phrased correctly.

According to the reviewer's comment, we changed this sentence as follows:

"In addition to histone acetylation, ATP-dependent cooperative chromatin remodelling can also change chromatin structure." (line 90-91)

- line 100, explain that ALC1 is an ATP-dependent chromatin remodeler, but it is not part of any of the four chromatin remodeling complexes mentioned in line 96

Thank you for your comment. We have included this explanation as follows:

"In addition, PARP1 associated with DDB2 facilitates polyADP-ribosylation of UV-damaged chromatin and recruits ALC1²⁴, which is an ATP-dependent chromatin remodeler but not part of any of the four chromatin remodeling complexes." (line 95-98)

- line 123, 'hardly detected' should be 'not detected' (unless the authors provide evidence that there is some accumulation'.

We agree with this comment. We have changed this wording. (line 116)

- overall, the authors should clearly indicate in each legend and figure the amount of UV damage induced and the time after UV at which the experiment was performed. This is not clear for all figures. In addition, it is not clear for all figures which cell lines were used. Are these in all cases HeLa, unless otherwise indicated? This should be made more clear.

We have now included the amounts of UV and the culture time after UV in the figure legends. We also clarified the cell lines used in analyses in all the figures.

- Lines 134-137. It is unclear why the authors speculate that HBO1 pS50/53 binding to damaged chromatin was decreased because of hydrochloric treatment, as they state that the sensitivity in MeOH/acetone treated samples is lower. Please clarify in the text.

As described above, we removed all co-localization graphs from the revised manuscript and therefore in the revised manuscript we do not mention the effect of hydrochloric treatment for pS50/53 staining. Our previous speculation was based on our judgment by eye. We would like to retract this speculation.

- line 231, 70% is incorrect judging from fig 3e.

We agree with this comment. We removed the previous Fig. 3e in this revised manuscript.

- line 236, one can hardly call co-localization of 30-40% 'abundant'. The co-localization level is slightly above shCtl (and similar to the shCtl of fig 3g!!).

We agree with this comment. We removed the previous Fig. 3e and g in this revised manuscript.

- line 287, IR induces oxidative damage, single strand and some double strand breaks. MMC induces both inter- and intrastrand crosslinks. Intra-strand crosslinks are repaired

by NER. This should be mentioned and can also explain the HBO1-dependent sensitivity the authors observe for MMC.

Thank you for this helpful comment. We have included this information as follows:

“Based on these observations, we examined whether HBO1 is involved in survival against ionizing radiation (IR), which induces oxidative damage, single strand breaks (SSBs) and some DSBs, or mitomycin C (MMC) treatment, which induces both inter- and intrastrand crosslinks.” (line 267-270)

“This is consistent with our observation that HBO1-depleted cells showed increased sensitivity for MMC, because MMC induced-interstrand crosslinks require the NER pathway to repair these types of DNA lesions⁴⁸.” (line 484-487)

- line 346, Figs 7A and S6C, if the authors wish to show that ACF1 (and SNF2H) do not respond to DSBs induced by UV irradiation, they should perform these local UV damage experiments in non-replicating cells. In replicating cells, UV damage can induce DSBs and/or replication stress that activates ATR, which subsequently phosphorylates H2AX. To observe that H2AX phosphorylation after UV is dependent on ATR, therefore does not prove that in the local damage shown in Figs 7A and S6C no double strand breaks or replication stress is induced. Also, the authors should indicate at which time point after UV these local damage experiments were performed.

We understand the reviewer’s suggestion. We removed the previous Supplementary Fig. 6c from the revised manuscript. To follow the reviewer’s recommendation, we performed local UV damage experiments in non-replicating cells. We performed synchronization of human fibroblasts at G0 by serum starvation as described in a previous report⁴⁹. We demonstrate accumulation of ACF1 at UV irradiated sites in **Supplementary Fig. 6a**. We have also indicated time points in the figure legend.

- line 428, HBO1 phosphorylation appears already saturated 30 min after UV in Fig 1b

We agree with this comment. We have presented this point in the text as follows:

“phosphorylation of HBO1 Ser50 and Ser53 on chromatin was already at maximum levels at 0.5 until 2 h after UV irradiation (Fig. 1b).” (line 416-417)

- line 434, maximum loading of XPC appears is already evident at 30 min after UV in Fig 1b.

We agree with this comment. We changed the text as follows:

“XPC loading on chromatin was at maximum levels from 0.5 h after UV irradiation (Fig. 1b).” (line 422-423)

- line 445-446, UV should not be listed as genotoxic agent that induces (only) DSBs. It would be better to speak of genotoxic agents that induce (single and double) strand breaks, oxidative damage, replication stress and helix-distorting DNA damage.

Thank you for your helpful suggestion. We have clarified this in the text as follows:

“depletion of ACF1 and SNF2H sensitizes cells to genotoxic reagents that induce (single and double) strand breaks, oxidative damage, replication stress and helix-distorting DNA damage^{26, 37}.” (line 432-434)

- English style and grammar could be improved throughout the manuscript

We submitted our manuscript for proofreading by a native English speaker and have included a certification document for reference.

Reviewers' comments:

Reviewer #2 (Remarks to the Author):

1. The authors have appropriately addressed this concern.
2. The authors have now consistently quantified relative fluorescence and explained how this was measured. Still, the way relative fluorescence was calculated is uncommon. Now the authors normalize signal intensity of a given protein at the site of damage by the signal intensity of the antibody used for damage staining (CPD or DDB2). It is good to take the amount of damage into account and thus to normalize by the intensity of the damage marker. However, it is also (more) common, to divide (normalize) average signal intensity at the site of damage by the average signal intensity of the protein throughout the nucleus (see for instance previous Nat Communications paper Puumalainen et al 2014). This way of normalization provides a fold increase of a protein at the site of damage compared to the overall levels of the protein in the remainder of the nucleus. We leave it to the editor to decide whether the current quantifications suffice. The authors now correctly discuss that ATR can be activated dependent on NER activity in G1 cells. However, this also occurs in S/G2 phase cells (besides ATR's activation by replication stress). It is therefore weird that the authors only find that HBO phosphorylation is affected by NER deficiency in G1 cells but not S/G2 cells.
5. The authors explain in length how they repeated this experiment in the right way and with a proper control (GFP-XPA). It is still unclear why in the first version of the manuscript, the authors found accumulation of SNF2H to 405 nm laser without Hoechst while they now show this cannot be repeated, while they still find, with the addition of Hoechst, that SNF2H accumulation depends on HBO1 and DDB2. We leave it to the editor to decide whether this is acceptable.
6. The authors have appropriately addressed this concern.
7. The authors have now appropriately performed the experiment shown in Fig 4B. It appears that HBO1 is necessary for the increase in H4Ac and H3AcK14 levels after UV. However, also without UV, HBO1 seems to regulate H3AcK14. This could be mentioned.
9. The authors have now performed these experiments with UV doses more common in literature.
10. The authors have appropriately addressed this concern.
12. The authors have appropriately addressed this concern.
14. The authors have addressed our concern by repeating the experiment and now show a clear effect of siSNF2H-2 on XPC-eGFP accumulation. We wonder whether this means that all live cell imaging experiments were only performed once or whether experiments were replicated to test for reproducibility. This is an important point to consider as the authors now also show in 'For Reviewer Figure 1' that one of their initial experiments (in the first version of the manuscript) with SNF2H could not be reproduced (see also point 5). We leave it to the editor to judge this.
17. This concern is addressed.

All minor concerns were addressed appropriately.

RESPONSES TO REVIEWER COMMENTS

1. The authors have appropriately addressed this concern.

2. The authors have now consistently quantified relative fluorescence and explained how this was measured. Still, the way relative fluorescence was calculated is uncommon. Now the authors normalize signal intensity of a given protein at the site of damage by the signal intensity of the antibody used for damage staining (CPD or DDB2). It is good to take the amount of damage into account and thus to normalize by the intensity of the damage marker. However, it is also (more) common, to divide (normalize) average signal intensity at the site of damage by the average signal intensity of the protein throughout the nucleus (see for instance previous Nat Communications paper Puumalainen et al 2014). This way of normalization provides a fold increase of a protein at the site of damage compared to the overall levels of the protein in the remainder of the nucleus. We leave it to the editor to decide whether the current quantifications suffice.

We agree that each method provides useful information. We have compared our calculation method with Puumalainen's method for the normalization of pS50/53 HBO1 and DDB2 co-immunostained cells as shown in Attached Fig. 1. We randomly chose four DMSO control HeLa cells and seven ATR inhibitor-treated HeLa cells and measured their relative pS50/53 HBO1 intensities to DMSO #1 using our own (Niida) and Puumalainen's methods, as shown in the legend. We calculated the mean of these relative intensities of ATR inhibitor-treated cells to DMSO control using each of the methods, as shown in Attached Fig. 1b. We found that the two methods provided very similar results for single cells (a) and combined cells (b). Other researchers have calculated the relative intensity of DDB2 normalized by CPD (Attached file "Wang et al.": Wang et al. 1722–1733 Nucleic Acids Research, 2013, Vol. 41, No. 3). Therefore, we would like to continue to use our calculation method in the manuscript.

The authors now correctly discuss that ATR can be activated dependent on NER activity in G1 cells. However, this also occurs in S/G2 phase cells (besides ATR's activation by replication stress). It is therefore weird that the authors only find that HBO phosphorylation is affected by NER deficiency in G1 cells but not S/G2 cells.

We understand that replication-independent ATR can be activated through the NER pathway in the G2 phase (Stiff et al. 3247–3253 Human Molecular Genetics, 2008, Vol. 17 No. 20

(Ref. 49)). However, we speculate that replication-dependent ATR activation is dominant compared with replication-independent–NER-dependent activation of ATR during the S phase. Furthermore, cells including replication-dependent activated ATR accumulate in the G2 phase after UV irradiation. If we could separate G2 cells to exclude cells with replication-dependent activated ATR, we might detect reduced phosphorylation of pS50/53 HBO1 in shXPC HeLa or XPA fibroblasts. However, we speculate that such a population is minor and it is impossible to separate replication-dependent activated ATR by our method. We discuss this point and cite the paper by Stiff et al. 2008 in the “Discussion” section of the revised manuscript (lines 482–487 and Ref. 49):

“We did not detect a significant reduction in HBO1 phosphorylation in S-G2/M shXPC and XP-A cells. However, replication-independent activation of ATR requiring NER was previously reported⁴⁹. We speculate that replication-dependent ATR activation is dominant in S phase cells and these cells accumulate in the G2 phase at the G2 checkpoint. Lack of replication-independent ATR activation in shXPC or XP-A cells could not be detected.”

5. The authors explain in length how they repeated this experiment in the right way and with a proper control (GFP-XPA). It is still unclear why in the first version of the manuscript, the authors found accumulation of SNF2H to 405 nm laser without Hoechst while they now show this cannot be repeated, while they still find, with the addition of Hoechst, that SNF2H accumulation depends on HBO1 and DDB2. We leave it to the editor to decide whether this is acceptable.

We failed to show irreproducible results regarding GFP-SNF2H in the previously submitted manuscript. Although we could not show reproducible GFP-SNF2H accumulation without Hoechst conditions in shCtl HeLa cells, we could show accumulation of GFP-SNF2H in shCtl HeLa cells and reduced accumulation in shDDB2 and shHBO1 HeLa cells with Hoechst conditions. Such experiments were performed independently at least three times. The results of these experiments are shown in **Attached Fig. 2**. We also show each independent result of GFP-SNF2H accumulation with Hoechst in **Attached Fig. 3**. We hope that this provides sufficient evidence of the effect of HBO1 on the accumulation of SNF2H in response to UV irradiation.

6. The authors have appropriately addressed this concern.

7. The authors have now appropriately performed the experiment shown in Fig 4B. It appears that HBO1 is necessary for the increase in H4Ac and H3AcK14 levels after UV. However, also without UV, HBO1 seems to regulate H3AcK14. This could be mentioned.

We agree. There are reports that HBO1 regulates H3AcK14 levels in DNA replication and transcription (Feng Y et al. and Kueh AJ et al., respectively). We have added a sentence on this point and cite these papers in the revised manuscript (lines 245–246 and Ref. 22 and 32):

“Consistent with previous reports^{22, 32}, HBO1 regulates H3AcK14 in the absence of UV damage.”

9. The authors have now performed these experiments with UV doses more common in literature.

10. The authors have appropriately addressed this concern.

12. The authors have appropriately addressed this concern.

14. The authors have addressed our concern by repeating the experiment and now show a clear effect of siSNF2H-2 on XPC-eGFP accumulation. We wonder whether this means that all live cell imaging experiments were only performed once or whether experiments were replicated to test for reproducibility. This is an important point to consider as the authors now also show in ‘For Reviewer Figure 1’ that one of their initial experiments (in the first version of the manuscript) with SNF2H could not be reproduced (see also point 5). We leave it to the editor to judge this.

We performed these experiments independently at least three times. We confirmed the reproducibility of our live cell imaging experiments except for GFP-SNF2H accumulation without Hoechst conditions, as mentioned above. In Supplementary Fig. S8, we show details of repeated live cell imaging experiments using siSNF2H-2-treated cells because we wanted to demonstrate depletion of endogenous SNF2H by siSNF2H-2 treatment decreased the accumulation of XPC-EGFP at damage sites. Consequently, we have data on XPC-EGFP accumulation in siSNF2H-2 cells from at least six independent experiments. We think that combining and statistically analyzing these data provides more accurate results. Therefore, we would like to include the results as new **Supplementary Fig. 8b**. We also provide each

independent result in **Attached Fig. 4** for the editor and reviewers.

17. This concern is addressed.

All minors concerns were addressed appropriately.